**Brief Communication**

# Chase-away evolution maintains imperfect mimicry in a brood parasite–host system despite rapid evolution of mimics

Tanmay Dixit [1,2] ✉, Jess Lund [1,2], Anthony J. C. Fulford[1], Andrei L. Apostol[3], Kuan-Chi Chen[3], Wenfei Tong[1], William E. Feeney [4,5], Lazaro Hamusikili[6], John F. R. Colebrook-Robjent[6,7], Christopher P. Town[3] & Claire N. Spottiswoode [1,2]

We studied a brood parasite–host system (the cuckoo finch *Anomalospiza imberbis* and its host, the tawny-flanked prinia *Prinia subflava*) to test (1) the fundamental hypothesis that deceptive mimics evolve to resemble models, selecting in turn for models to evolve away from mimics ('chase-away evolution') and (2) whether such reciprocal evolution maintains imperfect mimicry over time. Over only 50 years, parasites evolved towards hosts and hosts evolved away from parasites, resulting in no detectible increase in mimetic fidelity. Our results reflect rapid adaptive evolution in wild populations of models and mimics and show that chase-away evolution in models can counteract even rapid evolution of mimics, resulting in the persistence of imperfect mimicry.

Ever since Bates observed the remarkable similarity between different South American butterfly species[1], the phenomenon of mimicry has been used to illustrate how natural selection can produce striking adaptations. For mimicry to exist, mimics must evolve to resemble models. If models benefit from being discriminable from mimics, 'chase-away' selection should drive models to evolve away from mimics[2]. Thus, chase-away evolution could prevent the accuracy of mimetic resemblance, termed mimetic fidelity, from increasing over time[3,4]. However, few studies have examined evolutionary trajectories of both models and mimics simultaneously, probably because the required long-term data are difficult to obtain. To our knowledge, the only study to have examined changes in mimetic fidelity over time found that, for one of four traits studied, mimetic fidelity increased over time[5]. While this might suggest that chase-away selection was insufficient to prevent increases in mimetic fidelity over time, it is unknown whether the trait is used in discriminating between models and mimics and thus whether observed patterns were due to selection in the context of mimicry. Here, we study an aggressive mimicry system over 50 years to test the

hypothesis that chase-away selection on models prevents increases in mimetic fidelity over time.

The cuckoo finch *Anomalospiza imberbis* lays eggs which imperfectly mimic the complex and variable patterns of eggs of its host, the tawny-flanked prinia *Prinia subflava* (Methods), which reject mismatched eggs from their nests[6]. Individual prinias lay eggs with distinct colour and pattern phenotypes (egg signatures; Fig. 1a), such that a given cuckoo finch egg will be a poor match to most prinia clutches in the population[6]. Cuckoo finch eggs (mimics) exhibit simpler patterns than prinia eggs (models) and differences in pattern complexity predict egg rejection by prinias[7]. Egg rejection therefore has fitness consequences for both hosts and parasites and this implies that selection should favour parasites evolving towards hosts (that is, evolving increased complexity) and hosts evolving away from parasites (that is, also evolving increased complexity). By quantifying pattern complexity of 414 prinia and 162 cuckoo finch eggs from 1970 to 2020 (Methods), we tested whether host and parasitic phenotypes have changed in the predicted direction in the recent past and whether such reciprocal

[1]Department of Zoology, University of Cambridge, Cambridge, UK. [2]DST-NRF Centre of Excellence at the FitzPatrick Institute of African Ornithology, University of Cape Town, Rondebosch, South Africa. [3]Computer Laboratory, University of Cambridge, Cambridge, UK. [4]Behavioural and Evolutionary Ecology Group, Doñana Biological Station (CSIC), Seville, Spain. [5]Department of Biosciences, Durham University, Durham, UK. [6]Musumanene Farm, Choma, Zambia. [7]Deceased: John F. R. Colebrook-Robjent. ✉e-mail: td349@cam.ac.uk

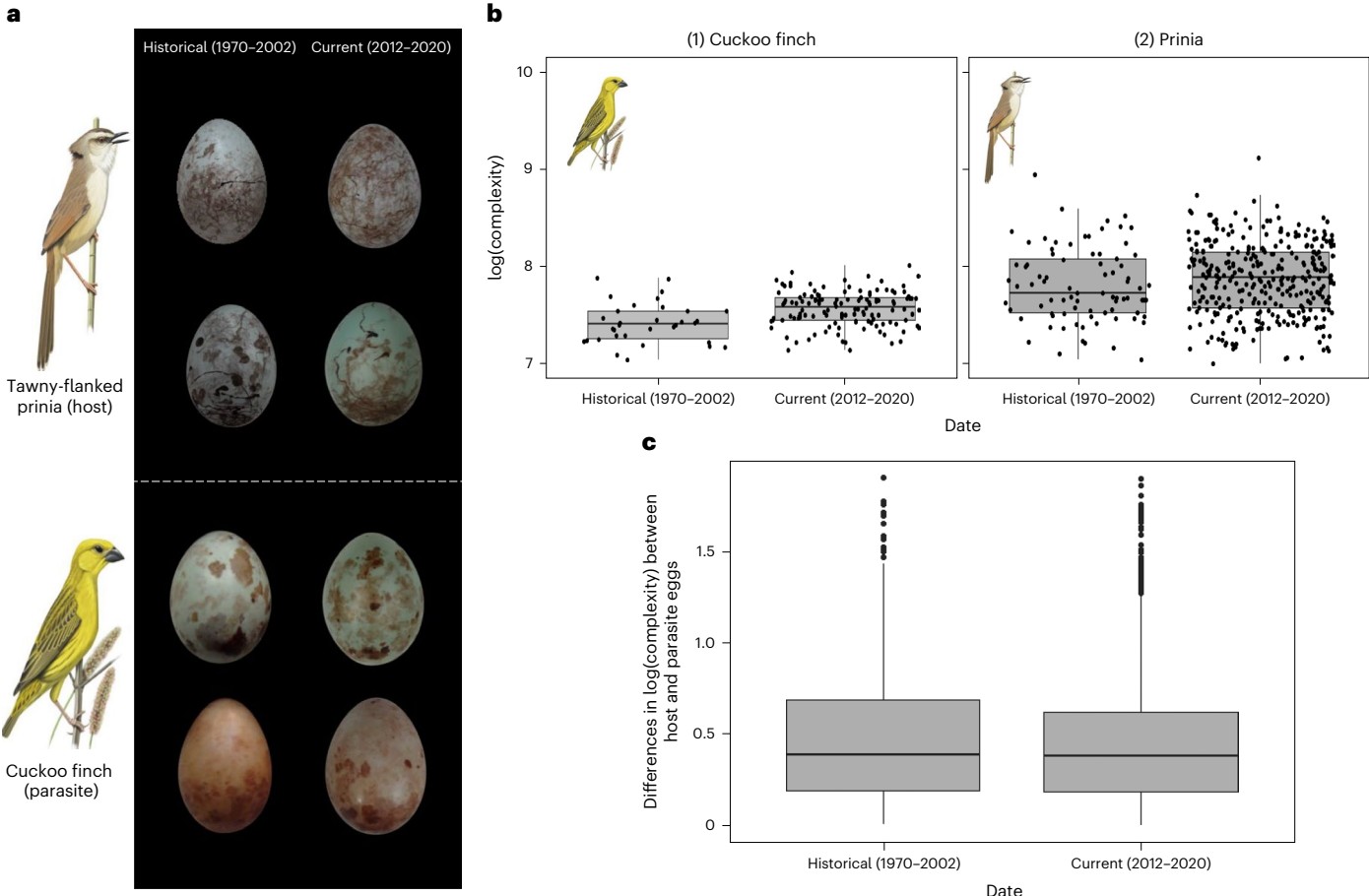

**Fig. 1 | Changes in egg pattern complexity over time. a**, Randomly selected host (above) and parasitic (below) eggs, from the historical (left) and current (right) samples. **b**, Changes in egg pattern complexity (log-transformed) over time in (1) parasites (*n* = 162 biologically independent eggs) and (2) hosts (*n* = 414 biologically independent eggs). Boxes range from the 25th percentile and horizontal lines represent medians. Minima and maxima are defined by the smallest datapoint no lower than the 25th percentile minus 1.5× interquartile range (IQR) and the largest datapoint no greater than the 75th percentile plus 1.5× IQR, respectively. All datapoints are shown as dots. **c**, Mimetic

fidelity through time: no significant change in differences in log(complexity) between all historical (*n* = 2,788) and current (*n* = 42,496) pairs of parasites and hosts These pairs were generated from the 162 parasitic eggs and 414 host eggs photographed. Boxes range from the 25th to the 75th percentile and horizontal lines represent medians. Minima and maxima are defined by the smallest datapoint no lower than the 25th percentile minus 1.5× IQR and the largest datapoint no greater than the 75th percentile plus 1.5× IQR, respectively. Outliers are shown as dots. Individual points are excluded as they obscure the boxplot. Bird illustrations reproduced with permission from faansiepeacock.com.

evolution led to any change in mimetic fidelity over time. We measured complexity (a synthetic measure of several pattern traits; Methods) on a logarithmic scale, since hosts perceive this measure of complexity according to Weber's Law[7]. Because effect sizes on logarithmic scales are not intuitive, we provide estimates as percentages where appropriate.

A linear model confirmed that in this dataset, prinia eggs are more complex than cuckoo finch eggs (estimate = 58%, 95% confidence interval (CI) = 30–93%, $t_{572}$ = 4.3, *P* < 0.001). Complexity across both species increased slightly but significantly over 50 years (estimated increase = 0.5%, 95% CI = 0.05–0.9%, $t_{572}$ = 2.0, *P* = 0.04; Fig. 1b and Extended Data Fig. 1). There was no significant difference between species in the rate of increase in complexity (interaction between species and year during which the egg was laid: estimate = −0.4%, 95% CI = −0.9–0.1%, $t_{572}$ = −1.4, *P* = 0.15). Because heteroscedasticity in the data (that is, hosts exhibiting higher variance in complexity than parasites, probably as a result of diversifying selection on host phenotypes[6,8]) may invalidate model inferences, we bootstrapped the linear model (Methods). As a further validation, we categorized eggs into those laid from 1970 to 2002 (historical; predominantly 1980–1990) and those from 2012 to 2020 (current). All results were consistent with the original model (Methods).

Overall, the finding that complexity increases over time suggests that parasites have evolved towards hosts and that hosts have evolved away from parasites at a similar rate.

If chase-away evolution in hosts occurred at a similar rate to parasite evolution, as implied above, then we would expect to see limited increases in mimetic fidelity despite rapid evolution of parasites. To quantify changes in mimetic fidelity, we calculated all host–parasite complexity differences from 1970 to 2002 (historical) and from 2012 to 2020 (current) (Methods). Bootstrapped estimates of historical and current mimetic fidelity showed considerable overlap (mean historical complexity difference on a logarithmic scale = 0.46, 95% CI = 0.38–0.55; mean current complexity difference = 0.42, 95% CI = 0.39–0.45; Fig. 1c). This corresponds to no significant increase in this trait-based measure of mimetic fidelity (bootstrapped estimated increase = 4%, 95% CI = −3–12%; Fig. 1c). We also independently estimated mimetic fidelity using a discriminant analysis based on complexity. The discriminant analysis for historical eggs correctly assigned 72% of eggs to the correct species (bootstrapped 95% CI = 63–80%). There was no significant difference between this and the performance of the discriminant analysis for current eggs (mean increase in mimetic fidelity = 2%; bootstrapped 95% CI = −10–13%), which assigned 70% of eggs to the correct species

(bootstrapped 95% CI = 61–78%). This echoes the result of comparing pairwise combinations of eggs: both measures of mimetic fidelity indicate that no observable increase in mimetic fidelity occurred, as expected given the lack of any significant difference between hosts and parasites in the rate of change of pattern complexity over time. Thus, chase-away selection driving host evolution away from parasites probably explains why mimicry of pattern complexity remains imperfect in this host–parasite system.

Although observed changes in complexity conformed to a priori predictions of coevolution, this study is correlational. We must therefore consider alternative explanations which could influence host and parasitic eggs in tandem, such as selection on egg pattern complexity from predation or climate. However, the main predators at our field site are snakes, which rely mostly on olfaction and infrared, and prinia nests are enclosed, limiting egg visibility at long range[8]. Climate change also appears unlikely to select for increases in complexity, since increased temperatures are likely to select for fewer pattern markings (which absorb more heat than unmarked eggshells)[9]. Complexity is highly correlated with the number of pattern markings and weakly correlated with pattern coverage[7]; thus, increased ambient temperatures due to climate change should select for reduced complexity, contrary to our findings.

In summary, tracking model and mimic phenotypes over 50 years showed that despite rapid evolution of parasites, there was no detectible increase in their mimetic fidelity to hosts. This suggests that the coevolutionary response in hosts was strong enough to prevent increases in mimetic fidelity and so supports the hypothesis that the persistence of imperfect mimicry can be explained by chase-away evolution in models[3,4].

## Methods

### Study species
At our study sites, on Semahwa and Musumanene Farms (around 16.74° S, 26.90° E) and surrounding areas in the Choma District of southern Zambia, the cuckoo finch currently parasitizes four cisticolid warbler species[10]. Of these four species, tawny-flanked prinias are the commonest, and have the most variable (and subjectively the most complex) egg patterns[11]. High interindividual variation in prinias (i.e. the presence of egg signatures) provides an effective defence against parasites, since egg signatures facilitate the rejection of mismatched eggs from host nests[6,12]. Although many egg signature traits may be important for egg rejection in this system[6], we focussed on complexity because quantifiable differences between hosts and parasites in this trait allow us to make clear predictions about the direction of evolution, namely that both should evolve towards higher complexity[7].

### Photography of eggs
In all analyses, one egg per photographed clutch was included (prinia $n = 414$; cuckoo finch $n = 162$; from 1970 to 2020), with a single image considered representative of the egg's phenotype. Images of eggs collected from 1970 to 2002 (from the private collection of J.F.R.C.-R., collected by J.F.R.C.-R. and L.H. and deposited in the Livingstone Museum, Zambia) were taken by C.N.S. Most of these eggs were from the 1980s. Images from 2013 were taken by W.E.F., C.N.S. and W.T.; images from 2014 were taken by W.T. and C.N.S.; images from 2018 to 2020 were taken by T.D.; all other images were taken by C.N.S. Although host and parasite eggs were also studied in 2007–2009[6], these years were excluded from this study because images from 2007 to 2009 were not comparable to other images taken (due to differences in scaling and normalization[7]). In a few years, some host eggs were not photographed or analysed due to a specific research focus on parasitic eggs, and host eggs were not routinely photographed owing to time constraints. Parasitic eggs can be reliably distinguished from host eggs by the absence of 'scribbles' of pigment on their shells, which hosts always exhibit[13]. Images were taken in linearized RAW format, in shade with either a Nikon D90 camera with a 60 mm Micro-Nikkor lens or a Fuji Finepix S7000 camera. For eggs collected from 1970 to 2002, a 17% grey card was used to normalize images. For all other eggs, two grey standard squares (N6.5 and N5; reflectance values 36.2% and 19.8%, respectively) of an X-rite ColorChecker Passport (X-Rite) were used to normalize images.

In all images except for those from 2018 to 2020, only 'one side' of each egg was photographed. In 2018–2020, eggs were photographed four times, rotating the egg through 90° around the long axis after each image, to maximize the amount of pattern photographed[7,14]. This produced images of 'sides' a, b, c and d, where a is opposite c and b opposite d. When determining historical changes, we used complexity values from only one side of eggs photographed in 2018–2020 (side a), rather than the average of a and c as used previously[7]. Complexity values for different sides of the egg are highly repeatable[7].

### Image analysis
We used the MICA toolbox[15] in ImageJ to normalize and scale images to 29 px mm$^{-1}$, 'cut out' (i.e. remove from the background) and mask (i.e. add an artificial black background to) eggs and produce greyscale images from the green channel. The green channel was used because it corresponds closely to the sensitivity of avian double cones, thought to be involved in pattern processing[16]. Pattern features were extracted using NATUREPATTERNMATCH (NPM)[17]. NPM detects and encodes local features (SIFT features) as 132-dimensional vectors, which loosely correspond to pattern markings. Complexity of the egg pattern was then calculated as in ref. 7. Briefly, six traits were measured: (1) the number of pattern features, (2) the variation in position of features on eggs, (3) the variation in the scale (size) of features, (4) the variation in the orientation of features, (5) the Redies change, a measure of how much intensity (brightness) changes across an image and (6) a measure of clustering tendency of features and within-cluster feature variation. All but trait (5) were based on features extracted using NPM. An optimization algorithm optimized the complexity metric (defined as a linear combination of these six traits) such that the absolute complexity difference between an experimental egg and the host clutch in which it was placed would best predict rejection of the experimental egg. For full details of this quantification, see ref. 7. Although this metric was based on present-day rejection data, we found evidence that selection has acted on host and parasite pattern complexity in the recent past (see main text). This implies that the complexity metric is not only relevant to current host rejection behaviour but also was relevant to rejection behaviour in the recent past.

Because perception conforms to Weber's Law[7] (that is, hosts perceive relative, rather than absolute, differences in complexity), we quantified pattern complexity on a logarithmic scale, with estimates of percentage changes in complexity calculated as exp(estimate).

One concern with using historical egg collections is that the background colour of eggs can fade over time, especially if they are poorly stored, which was not the case for the eggs photographed as part of this study. Old eggs were photographed in 2007 and 2009 and eggs were kept in a darkened room and collected relatively recently[8]. Furthermore, it is largely blue-green colours on eggs which fade (for example, ref. 18), which has no relevance to the pattern measures we extracted, since pattern measures extracted from NPM should be unaffected by the underlying colour. In the unlikely event that fading affected the detectability of faint markings by NPM, any background colour fading on old eggs would make faint markings more detectable on these eggs, resulting in higher complexity scores for old eggs than for fresh eggs. Our results run counter to this (see main text) and are therefore conservative.

A second concern with studying host and parasitic egg phenotypes more generally is that some (probably poorly matched) parasitic eggs may be rejected from host nests before data from that nest are collected. This may mean that only closely matched parasitic eggs

are phenotyped. However, in this system this is unlikely to be a problem, since (1) hosts often take 1–4 days to reject a poorly matched egg (particularly eggs that are poorly matched in terms of pattern, rather than colour)[7] and (2) high variation in host egg appearance between clutches (Fig. 1b) means that all cuckoo finch eggs are poor matches to most of the host population at any given time. Thus, there is unlikely to be a bias towards phenotyping well-matched eggs.

### Testing for changes in complexity over time

All statistical analyses were conducted in R (v.4.0.2; ref. 19). We used linear models (function lm) to quantify change in complexity over time across species, using the model complexity ~ species + year + species : year. For example, a negative coefficient for the interaction term, with positive coefficients for the species and year terms would indicate that prinia complexity was greater than cuckoo finch complexity and that complexity increased over time but cuckoo finch complexity increased more than prinia complexity.

First, we tested for continuous changes over time. Year was modelled as a continuous variable with years assigned integer values from 0 (year 1970) to 50 (year 2020). Because prinias exhibited much greater variance than cuckoo finches (thus falsifying the model assumption of homoscedasticity), we bootstrapped the model to calculate 95% CI for model coefficients using 1,000 replicates. Results from bootstrapping were consistent with the initial model. CIs for the interaction term spanned zero (estimate = −0.4%, 95% CI = −0.7–0.003%), while CI for species (estimate = 58%, 95% CI = 35–86%) and year (estimate = 0.5%, 95% CI = 0.2–0.7%) did not span zero, indicating that prinia eggs are more complex than cuckoo finch eggs and that complexity increased over time. Median complexity appeared to fluctuate during certain periods (Extended Data Fig. 1), such that the overall increase in complexity was not monotonic. These fluctuations are probably due to low sample sizes of eggs from specific years, combined with very high population-wide variation in complexity. Such sampling error is especially likely during periods such as the mid- to late-1980s, in which sample sizes were low for each year; correspondingly, fluctuations in complexity were apparent in these years. However, with these data we cannot rule out other selective pressures or environmental influences driving short-term increases or decreases in complexity in one or both species. Regardless of the cause of these apparent fluctuations, they mean that we did not observe a monotonic increase in complexity in either species. Therefore, we conducted further analyses to test the robustness of the results of the linear model.

We subdivided the datasets into historical eggs (from 1970 to 2002; prinia $n = 82$, cuckoo finch $n = 34$) and current eggs (from 2012 to 2020; prinia $n = 332$, cuckoo finch $n = 128$). Results for this model were also consistent with the previous models (species−estimate = 34%, 95% CI = 23–45%, $t_{572}$ = 8.4, $P < 0.001$; year−estimate = 16%, 95% CI = 2–32%, $t_{572}$ = 2.3, $P = 0.02$; interaction−estimate = −9%, 95% CI = −22–7%, $t_{572}$ = −1.3, $P = 0.2$). Conclusions also remained unchanged when this model was bootstrapped (Species−estimate = 34%, 95% CI = 27–42%; year−estimate = 16%, 95% CI = 7–26%; interaction−estimate = −9%, 95% CI = −19–3%).

In summary, complexity was higher in prinias than cuckoo finches, complexity increased over time across both species and there was no detectable difference between species in the rate of increase of complexity over time.

### Testing for changes in mimetic fidelity over time

We measured mimetic fidelity using two methods. The first method calculated all host–parasite differences in each time period. Comparing all host–parasite pairs assumes that cuckoo finches lay their eggs at random in prinia nests; that is, independently of the patterning on the prinia eggs they contain. This has been shown to be a valid assumption in this system[6]. All possible pairings of parasite and host eggs measured at each time point ($n = 82 \times 34 = 2,788$ for historical

data; $n = 332 \times 128 = 42,496$ for current data) provide a sample from the joint distribution of parasite and host pairs in the population at large during each time period. To test whether mimetic fidelity had changed over time, we generated absolute differences in the logarithm of complexity between all pairs of host and parasitic eggs. From these we calculated the mean absolute difference for each time period and used a two-sample bootstrap with 500 replicates to estimate its 95% CI (mean ± 2 × bootstrap s.e.m.). The bootstrap was necessary because the number of paired differences contributing to the mean estimate far exceeded the available degrees of freedom: for instance, the sample size for current data ($n = 42,496$) was generated from 460 observations (d.f. = 459). Thus, simply conducting statistical tests without a bootstrap would overestimate statistical power, whereas bootstrapping allows calculation of confidence intervals which do not overestimate statistical power.

As a second measure of mimetic fidelity, we used flexible discriminant analysis (FDA; function fda in the R package mda[20]) with log(complexity) as the only predictor and with uninformed (that is, equal and unbiased) priors. A high-performing FDA would indicate low mimetic fidelity because high performance would imply that the algorithm can accurately assign eggs to species. Since the performance of an FDA tends to increase with sample size, we resampled current eggs to the same number as historical eggs ($n = 82$ prinia and $n = 34$ cuckoo finch). We ran 1,000 iterations of the FDA for both historical and current populations to calculate confidence intervals.

The two measures of mimetic fidelity used here correspond to slightly different questions. The mean host–parasite pairwise distance is an estimate of the average similarity (in terms of complexity) of a randomly selected host–parasite pair, for the historical and present-day subsets. This simulates the visual information available to guide the behaviour of a host female, who must compare her own egg(s) with the egg of a parasite. The FDA provides an estimate of the likelihood of assigning eggs correctly to species based on their complexity, for each subset. This considers whether mimetic fidelity has changed at a population level.

### Reporting summary

Further information on research design is available in the Nature Portfolio Reporting Summary linked to this article.

## Data availability

All data are available at https://doi.org/10.17863/CAM.101483 (ref. 21).

## Code availability

All code is available at https://doi.org/10.17863/CAM.101483 (ref. 21).

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

## Acknowledgements

In Zambia, we thank many people who assisted with nest-finding and fieldwork, including C. Moya, S. Hamama, S. Mwanza, S. Mukonko, O. Siankwasiya and C. Siankwasiya. We thank R. Duckett, V. Duckett, A. Sejani, T. Nicolle, E. Nicolle, I. Bruce-Miller and E. Bruce-Miller for allowing us to conduct fieldwork on their properties. We thank M. Greenshields and A. Greenshields for their hospitality. We also thank L. Chama, M. Chibesa and S. Siachoono at Copperbelt University for their support. We thank the Department of National Parks and Wildlife and the Zambia Wildlife Authority for permission to conduct research in Zambia. Research was conducted under permit no. DNPW/8/27/1. We thank M. C. Attwood, M. K. Dixit and G. A. Jamie for comments or discussion. T.D. was funded by a Balfour studentship from the Department of Zoology, University of Cambridge. C.N.S. was funded by a BBSRC David Phillips Fellowship (BB/J014109/1), a Royal Society Dorothy Hodgkin Fellowship and the DST-NRF Centre of Excellence at the FitzPatrick Institute, University of Cape Town. W.T. was funded by a BBSRC David Phillips Fellowship (BB/J014109/1) to C.N.S.

## Author contributions

T.D. and C.N.S. conceived of and designed the study. T.D., W.E.F., W.T., L.H., J.F.R.C.-R. and C.N.S. collected the data. A.L.A., K.-C.C. and C.P.T. designed methods for computation of complexity. A.L.A. conducted the computation under the supervision of C.P.T. and T.D. T.D., J.L. and A.J.C.F. processed and analysed the data. T.D., J.L., C.N.S. and A.J.C.F. interpreted the results. T.D. drafted the manuscript and C.N.S., J.L., K.-C.C., A.J.C.F., W.E.F., W.T. and C.P.T. contributed to later versions.

## Competing interests

The authors declare no competing interests.

## Additional information

**Extended data** is available for this paper at https://doi.org/10.1038/s41559-023-02232-4.

**Correspondence and requests for materials** should be addressed to Tanmay Dixit.

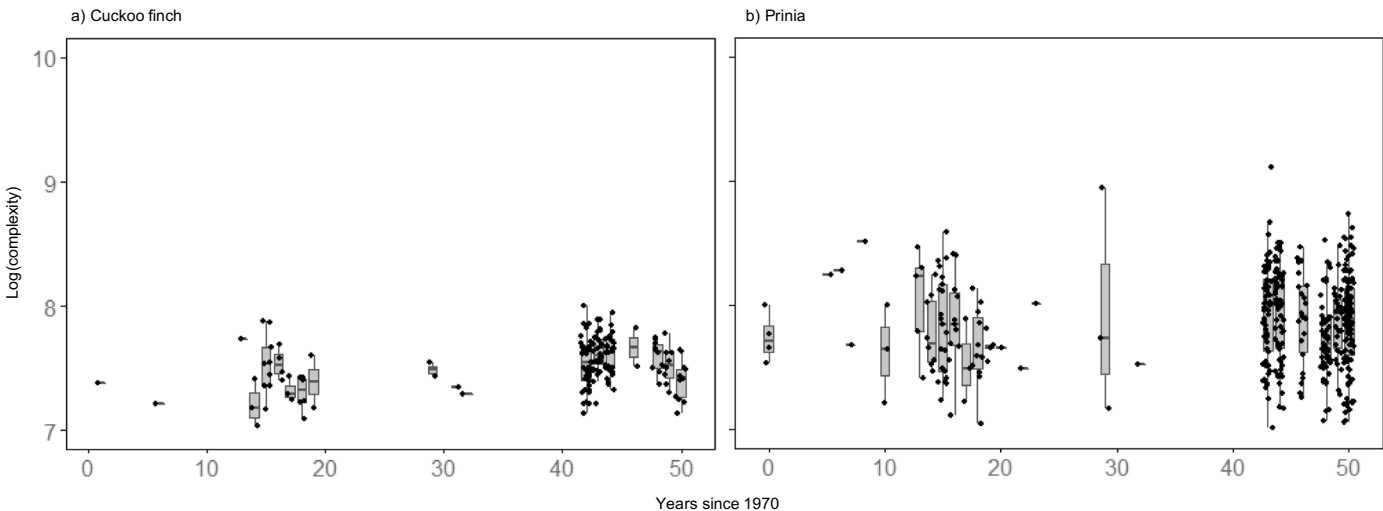

**Extended Data Fig. 1 | Changes in complexity across 50 years in cuckoo finches and prinias.** Change in complexity over time in **(a)** cuckoo finches and **(b)** prinias. Years are defined as number of years after 1970 (such that Year 0 corresponds to 1970 and Year 50 to 2020).

# Reporting Summary

## Statistics

For all statistical analyses, confirm that the following items are present in the figure legend, table legend, main text, or Methods section.

| n/a | Confirmed | |
|---|---|---|
| ☐ | ☒ | The exact sample size (*n*) for each experimental group/condition, given as a discrete number and unit of measurement |
| ☐ | ☒ | A statement on whether measurements were taken from distinct samples or whether the same sample was measured repeatedly |
| ☐ | ☒ | The statistical test(s) used AND whether they are one- or two-sided *Only common tests should be described solely by name; describe more complex techniques in the Methods section.* |
| ☐ | ☒ | A description of all covariates tested |
| ☐ | ☒ | A description of any assumptions or corrections, such as tests of normality and adjustment for multiple comparisons |
| ☐ | ☒ | A full description of the statistical parameters including central tendency (e.g. means) or other basic estimates (e.g. regression coefficient) AND variation (e.g. standard deviation) or associated estimates of uncertainty (e.g. confidence intervals) |
| ☐ | ☒ | For null hypothesis testing, the test statistic (e.g. *F*, *t*, *r*) with confidence intervals, effect sizes, degrees of freedom and *P* value noted *Give P values as exact values whenever suitable.* |
| ☒ | ☐ | For Bayesian analysis, information on the choice of priors and Markov chain Monte Carlo settings |
| ☒ | ☐ | For hierarchical and complex designs, identification of the appropriate level for tests and full reporting of outcomes |
| ☐ | ☒ | Estimates of effect sizes (e.g. Cohen's *d*, Pearson's *r*), indicating how they were calculated |

*Our web collection on statistics for biologists contains articles on many of the points above.*

## Software and code

Policy information about availability of computer code

| Data collection | No software was used to collect data in this study. |
|---|---|
| Data analysis | All analyses were conducted in R version 4.0.2 |

For manuscripts utilizing custom algorithms or software that are central to the research but not yet described in published literature, software must be made available to editors and reviewers. We strongly encourage code deposition in a community repository (e.g. GitHub). See the Nature Portfolio guidelines for submitting code & software for further information.

## Data

Policy information about availability of data

All manuscripts must include a data availability statement. This statement should provide the following information, where applicable:
- Accession codes, unique identifiers, or web links for publicly available datasets
- A description of any restrictions on data availability
- For clinical datasets or third party data, please ensure that the statement adheres to our policy

All data are available at https://doi.org/10.17863/CAM.101483.

# Research involving human participants, their data, or biological material

Policy information about studies with human participants or human data. See also policy information about sex, gender (identity/presentation), and sexual orientation and race, ethnicity and racism.

| | |
|---|---|
| Reporting on sex and gender | *Use the terms sex (biological attribute) and gender (shaped by social and cultural circumstances) carefully in order to avoid confusing both terms. Indicate if findings apply to only one sex or gender; describe whether sex and gender were considered in study design; whether sex and/or gender was determined based on self-reporting or assigned and methods used. Provide in the source data disaggregated sex and gender data, where this information has been collected, and if consent has been obtained for sharing of individual-level data; provide overall numbers in this Reporting Summary. Please state if this information has not been collected. Report sex- and gender-based analyses where performed, justify reasons for lack of sex- and gender-based analysis.* |
| Reporting on race, ethnicity, or other socially relevant groupings | *Please specify the socially constructed or socially relevant categorization variable(s) used in your manuscript and explain why they were used. Please note that such variables should not be used as proxies for other socially constructed/relevant variables (for example, race or ethnicity should not be used as a proxy for socioeconomic status). Provide clear definitions of the relevant terms used, how they were provided (by the participants/respondents, the researchers, or third parties), and the method(s) used to classify people into the different categories (e.g. self-report, census or administrative data, social media data, etc.) Please provide details about how you controlled for confounding variables in your analyses.* |
| Population characteristics | *Describe the covariate-relevant population characteristics of the human research participants (e.g. age, genotypic information, past and current diagnosis and treatment categories). If you filled out the behavioural & social sciences study design questions and have nothing to add here, write "See above."* |
| Recruitment | *Describe how participants were recruited. Outline any potential self-selection bias or other biases that may be present and how these are likely to impact results.* |
| Ethics oversight | *Identify the organization(s) that approved the study protocol.* |

Note that full information on the approval of the study protocol must also be provided in the manuscript.

# Field-specific reporting

Please select the one below that is the best fit for your research. If you are not sure, read the appropriate sections before making your selection.

☐ Life sciences ☐ Behavioural & social sciences ☒ Ecological, evolutionary & environmental sciences

For a reference copy of the document with all sections, see nature.com/documents/nr-reporting-summary-flat.pdf

# Ecological, evolutionary & environmental sciences study design

All studies must disclose on these points even when the disclosure is negative.

| | |
|---|---|
| Study description | This study measured the pattern complexity of 414 tawny-flanked prinia and 162 cuckoo finch eggs from 1970–2020, testing whether complexity differed between the species and whether and how complexity had changed over time. |
| Research sample | The research sample was images taken of 414 tawny-flanked prinia and 162 cuckoo finch eggs from 1970–2020, all taken on Semahwa and Musumanene Farms (around 16.74'S, 26.90'E) and surrounding areas in the Choma District of southern Zambia. These were all images available of eggs of these species, and all images were taken largely to test other hypotheses. Images of eggs collected from 1970–2002 (from the private collection of JFRCR, collected by JFRCR and LH, and deposited in the Livingstone Museum, Zambia) were taken by CNS. Most of these eggs were from the 1980s. Images from 2013 were taken by WEF, CNS, and WT; images from 2014 were taken by WT and CNS; images from 2018–2020 were taken by TD; all other images were taken by CNS. [All abbreviations refer to authors in the author list.] |
| Sampling strategy | All available images were used. We used bootstrapping to confirm that results were not unduly influenced by sample size. |
| Data collection | Images were taken in linearised RAW format, in shade with either a Nikon D90 camera with a 60 mm Micro-Nikkor lens or a Fuji Finepix S7000 camera. For eggs collected from 1970–2002, a 17% grey card was used to normalise images. For all other eggs, two grey standard squares (N6.5 and N5; reflectance values 36.2% and 19.8% respectively) of an X-rite ColorChecker Passport (X-Rite, MI, USA) were used to normalise images. Nests were located by a team of nest-finders; one egg per clutch was used in any analysis to avoid pseudoreplication. |
| Timing and spatial scale | Data were collected from 1970–2020, with periods of peak collection in the 1980s and 2010s, and several years with no collection, particularly years during the 1990s and 2000s. Data were not collected with this study in mind, and therefore we could only rely on data that had been collected for other purposes, hence the uneven temporal distribution of data. Data were collected from an area of approximately 3000 hectares, including on Semahwa and Musumanene Farms (around 16.74'S, 26.90'E) and surrounding areas in the Choma District of southern Zambia. |
| Data exclusions | One egg per clutch was randomly selected to be used in analyses, in order to avoid pseudoreplication. |

| | |
|---|---|
| Reproducibility | This study did not involve experimental findings. We conducted bootstrapped analyses, which allowed us to calculate 95% confidence intervals. |
| Randomization | Randomization is not relevant to this study as it was not an experimental study and there was no assignment to groups. |
| Blinding | All data were collected without knowledge of the hypotheses (i.e., all egg images were taken for other purposes, and there was no bias towards photographing eggs of any particular pattern complexity). The hypotheses were generated prior to data analysis. |

Did the study involve field work?  ☒ Yes  ☐ No

# Field work, collection and transport

| | |
|---|---|
| Field conditions | Fieldwork was conducted on farmland and adjacent land in Zambia. |
| Location | Data were collected from an area of approximately 3000 hectares, including on Semahwa and Musumanene Farms (around 16.74'S, 26.90'E) and surrounding areas in the Choma District of southern Zambia. |
| Access & import/export | Analyses involved photographed eggs either from a historical egg collection (now housed in the Livingstone Museum, Zambia) or from fieldwork in which eggs were returned to nests after photography. No samples were exported. Fieldwork was conducted under permits from the Zambian Department of National Parks and Wildlife (DNPW; previously Zambian Wildlife Authority) under permit number DNPW/8/27/1. |
| Disturbance | No disturbance was caused when photographing eggs from the historical egg collection, as these had already been removed from natural settings. When eggs were - more recently - photographed in the field, this process was conducted quickly (<10 minutes for a clutch) to minimise any disturbance to the nesting birds. Neither species studied are of conservation concern. |

# Reporting for specific materials, systems and methods

We require information from authors about some types of materials, experimental systems and methods used in many studies. Here, indicate whether each material, system or method listed is relevant to your study. If you are not sure if a list item applies to your research, read the appropriate section before selecting a response.

## Materials & experimental systems

| n/a | Involved in the study |
|---|---|
| ☒ | ☐ Antibodies |
| ☒ | ☐ Eukaryotic cell lines |
| ☒ | ☐ Palaeontology and archaeology |
| ☒ | ☐ Animals and other organisms |
| ☒ | ☐ Clinical data |
| ☒ | ☐ Dual use research of concern |
| ☒ | ☐ Plants |

## Methods

| n/a | Involved in the study |
|---|---|
| ☒ | ☐ ChIP-seq |
| ☒ | ☐ Flow cytometry |
| ☒ | ☐ MRI-based neuroimaging |

