## [Peer Review File · Nature Ecology & Evolution]

Peer Review Information

Journal: Nature Ecology & Evolution

Manuscript Title: Chase-away evolution maintains imperfect mimicry in a brood parasite–host system despite rapid evolution of mimics.

Corresponding author name(s): Tanmay Dixit

Editorial Notes:

Reviewer Comments & Decisions:

Decision Letter, initial version:
--

5th January 2023

Dear Mr Dixit,

Your Brief Communication entitled "Rapid evolution of a brood parasite's egg pattern does not lead to large increases in mimetic fidelity." has now been seen by three reviewers, whose comments are attached. In the light of their advice, we have decided that we cannot offer to publish your manuscript in Nature Ecology & Evolution.

From the reports, you will see that while they find your work of some potential interest, reviewer 1 raises concerns about the advance your findings represent over earlier work (particularly <https://academic.oup.com/biolinnean/article/121/1/50/2999302> and your co-authors' previous work, for example citation 8) and the strength of the novel conclusions that can be drawn at this stage. We feel that these criticisms are sufficiently important as to preclude publication of your work in Nature Ecology & Evolution.

I am sorry that we cannot be more positive on this occasion, but hope that you find the reviewers' comments helpful when preparing your paper for resubmission elsewhere.

[REDACTED]

Reviewer expertise:

Reviewer #1: brood parasite ecology

Reviewer #2: signed report

Reviewer #3: brood parasite ecology

Reviewers Comments:

Reviewer #1 (Remarks to the Author):

There is already a prior paper on this topic, that spans a 100 year long period, and shows that host-parasite mimicry co-evolved towards better match over the time: Geltsch et al. 2017 BJLS. You'll need to please consider in what way your result is novel and not incremental or specific to your study system.

I am also wondering about color--color plays a critical role in your study systems' mimicry but it is not considered in my judgement here fully in this paper, even though the senior author has already published on the pattern (and some lack of) coevolution in color between this host and parasite in the early 2010s.

I would like to see Fig. 1b represent the linear analyses, not the historical vs. modern comparisons, since the methods differed between the collections of these eggs/samples and we know that collection specimens fade a lot over years/decades compared to fresh eggs from the field. How did you correct for that factor or is the trait set that goes into your complexity metric not affected by storage over time?

Reviewer #2 (Remarks to the Author):

Dear Tanmay et al.,

I greatly enjoyed your article "Rapid evolution of a brood 1 parasite's egg pattern does not lead to large increases in mimetic fidelity." In this article, you explored pattern complexity in a population of an avian brood parasite and its host using a 50-year long-term dataset. Unlike many, you did not assume that the hosts' eggshell phenotypes were static. Instead, you considered whether reciprocal selection pressures on the parasite and host populations may alter eggshell phenotypes in both, and if so, how this might alter the fidelity of eggshell mimicry. You found that the tawny-flanked prinias' (hereafter, prinia) eggs were more complex than those of cuckoo finches, that eggshell complexity increased (slightly) over time, and that the level of mimetic fidelity appeared consistent over time.

While I do have suggestions for alternative approaches, it seems the data and the patterns they illustrate are clear enough. In my opinion, the strength of this article is that it considers both host and parasite eggshell phenotypes. In most cases, coevolutionary arms races are assumed and only the egg traits of the parasites and the egg recognition abilities of the hosts are considered. Articles such as this, will help break us from that mold and yield much more insightful arguments about host-

2parasite dynamics. I hope you find my comments useful in revising your manuscript.

With best wishes,

Daniel Hanley

Major comments:

You have initially framed your paper around the premise that models evolve away from mimics, and host (rapid) evolution can counteract the eggshell adaptations of their parasites and yield imperfect mimicry. My first three comments will relate to these points, but more general comments will follow.

1. Models evolving away from mimics: In this manuscript you measured eggshell complexity in both hosts and parasites. You found that hosts were more complex than parasites and there was a slight increase in complexity over time, but there was no significant interaction between species (host or parasite) and time. First, a particular level of complexity does not imply the same actual pattern on the host and the parasite. Second, I didn't recall seeing a test that showed parasites evolving "toward" hosts or hosts "away" from parasites (see below). Had the interaction been significant, such that complexity of the parasites' eggs were evolving to be more similar to that of the host, then perhaps these statements would be justified. While I understand that you are presented evidence that is consistent with this assumption (e.g., Line 66), it seemed conspicuously odd that you didn't actually test whether hosts were evolving 'away' from mimics directly. Please see my suggestions for other approaches below.

2. Rapid evolution of parasites: You referred to rapid evolution repeatedly (title and lines 26, 28, 69, and 88); however, I saw no evidence of this. Your work focused on trait complexity, rather than particular colors and patterns. The changes in complexity over time were modest (and not particularly well described linearly – see suggestions below), and you found no appreciable difference in the fidelity of mimicry. So, as the result, I struggled to see what rapid evolution you were talking about. I do understand that in some of these cases you were speaking more generally, that rapid evolution in one population could counteract the evolution in another; however, it did seem that you were stating the parasites were rapidly evolving and their hosts were managing to 'keep their distance.' I might remove 'rapid' from the title and include something instead about the evolutionary process itself (e.g., something with "chase").

3. Imperfect mimicry: An apparent main conclusion is that chase-away evolution can explain the maintenance of imperfect mimicry. You raised this as an important point in the abstract (line 28) and reintroduced the idea later in the main text (~line 86). While this may be true, I think this point should take a less prominent role. I think that this model is much more powerful as an "alternative" process, and that more emphasis is needed on the evolutionary processes (rather than the consequence). For example, a chase-away model does not require constant fidelity of mimicry, as mimicry can improve and worsen over time via this process (see suggestions below), what is most important is the processes and dynamic between the host and parasite. I was expecting more emphasis on alternative processes (e.g., red queen dynamics) and less on the particular (null) pattern. To be fair, your article was quite short, so I understand if you think it isn't quite fair for me to

3say that this took a “prominent role.” I also realize that the fidelity of mimicry illustrates the evolutionary process, which I’m arguing is your main point. However, because the paper was so short, it also had a few easily digestible take-home messages. For me, this was one (important) one, and will be a useful vehicle for arguing chase-away evolution (since the hosts ability to stay differentiated is evidence of the process). I think it may be useful to consider a slight shift in emphasis. Again, apologies for subtlety here, but it seems that the constant fidelity of mimicry (currently over-emphasized) is evidence of the chase-away evolution (currently under-emphasized).

As a secondary point relating to “imperfect mimicry,” I think it would be beneficial to make it clearer that the complexity of the eggshell patterns are not perfectly matched. Your study species has nearly perfect eggshell color mimicry in several distinct morphs. While individual host females tend to lay eggs of these similar colors, each has distinct eggshell patterns. Thus, while a cuckoo-finch female may be able to match the colors/pigments for a broad subset of the host population, she will not be able to successfully produce a pattern that matches those same females. Therefore, for those specific features (eggshell patterns) we expect imperfect mimicry. Those patterns would be selected by multiple host females (with differing features) and likely generate some intermediate phenotype that is a reasonable facsimile to those found in the host population. These concepts are only quickly introduced (lines 44-47), but I suspect that the importance of these lines will be lost on most readers.

4. Analysis: Your data are not particularly continuous. Instead, you have data from two main time periods. Over this time series, you (naturally) have more data on host eggs, which also (naturally) have more variability. You have provided several analyses to overcome the challenges these data might impose, and I think that all were interesting, informative, and it would appear well executed. However, I do struggle to understand why complexity would be linearly related to “year” or how that might relate to your underlying hypothesis. Instead, I think that you actually should analyze how well the parasite population “tracks” your host population over time. This would still allow you to demonstrate the fidelity is constant because both the host and parasite population vary complexity over time. The conclusions that complexity is greater in more recent years is challenging to accept when looking at the data, as it seems that there are increases and decreases (as one would expect with chase-away evolution) over time. Moreover, host and parasite population seem to track one another. From this perspective a temporal lag, would actually suggest that hosts are evolving away from mimics (or mimics are evolving toward models). These models do not need to be overly complex, there is a rich tradition in population ecology for models that compare two populations over time.

5. Alternatives: I would appreciate more alternative explanations. For example, red queen dynamics seem like a feasible explanation of the observed patterns. Are there functional or practical differences between these hypotheses? Could they both apply but have slightly different foci? Moreover, if you examine the differences between these populations over time (see the previous suggestions) would seasonal differences in rainfall or diet easily explain the apparent increases and decreases in complexity? Are eggs more complex when they have more surface pigments? Aviles et al. 2017 did find season changes in eggshell pigmentation that related to rainfall, and (at least visually) increases and decreases of complexity appear to also track with ENSO patterns (fluctuations in El Niño and La Niña events). For the record, I’m not asking you to write a paper about climate and eggshell colors; instead, I present this as an example of an alternative that might explain why both populations shift in their eggshell features consistently over time. Overall, I would appreciate it if you could give the

4alternatives greater attention.

Minor comments:

Line 43: It is great that you have long-term data, and these data should be promoted. However, in reality, your dataset has two periods of active collection. While this has some statistical implications, I think this suggests you should exhibit some caution on such claims.

Line 50: This is a truly fantastic time series. I would presume these are relatively evenly distributed through time? For example, ~8 host eggs each year (but I'm not asking for these details here). Upon closer inspection, it is clear that you have good sampling in only a few of these years.

Line 54: It isn't clear to me how you are presented complexity (log scale) as a percentage. Perhaps another approach would be to keep the metrics in their units but discuss their effect sizes as odd ratios or similar metrics.

Line 56: Why would complexity have increased significantly over the last 50 years unless you caught the hosts and parasites early in their interaction? A chase-away model would imply that complexity - changes- not necessarily gets more complex. A closer inspection of your supplemental figure shows that this is the case, the complexity fluctuates in both populations repeatedly throughout the time series. This calls into question any expected linear differences between complexity and time. In fact, it isn't clear why this is an expected relationship under the chase-away model at all (when the hypothesis simply states that the model should evade the mimic). This requires more explanation.

Line 56: I would prefer for these data to remain in the log scale with the appropriate CI. These details will be in your methods, and are useful for those reading the statistics within the parentheses. You could/should present these as odd ratios for a more intuitive way to present the information to all readers.

Line 61: Is the greater variance in the host population actually a feature of the chase-away model, or simply a statistical (/practical) artefact?

Line 66: It may be my preference, but I do not think that the 1 and 2 are really necessary. The sentence and take-away message are simple enough without them.

Line 68: In my opinion, you may benefit from merging the content from current lines 68-75 on line 60 (just before the new sentence). You could likely condense your alternative test explanations (62-64) and then focus more on interpretation, which felt a little lacking and lines 65-67 felt premature. They would have much more weight after your full presentation and strengthen your last paragraph or set it up, the last sentence of your penultimate sentence. If you choose to follow this advice, you may need to rework some of the text a little.

Lines 73-75: You are swapping between log scales and percentages here. Also, the difference between the two log scales is 0.04. However, unless I'm missing something (which is entirely possible) a 0.04 difference is not the same thing as a 4% increase. Finally, you state that there is "no significant" increase, but the overlap is greater for your historical data. So, shouldn't that be "This [does not

5represent a] [...] significant [decrease] in this trait-based measure of..." I've added brackets for added, removed, or altered words and I think that this section could be more carefully worded. Additionally, I find this a very confusing presentation. Instead, I strongly recommend illustrating the two bootstrapped distributions. It would be much more intuitive to show those and describe the overlap.

Line 73: This is a stylistic preference (ignore as you see fit - or follow the journal's suggestions). For your confidence intervals I like to present these as 0.38 to 0.55. Presenting them like this would avoid the curly brackets which are awkward, especially when alongside the parentheses.

Line 75: I suggest a slightly different presentation here. I would start by describing the overlap in mimetic fidelity and then the performance of your discriminant analysis to differentiate parasite eggs from host eggs. Presenting these as two complementary tests, rather than a primary and follow-up test would be useful here.

Line 76: I appreciate that you are providing an alternative approach (and I like the approach you used); however, do you have any reason to assume this is a false negative? The reasons would be useful here. Depending on your reason, it may also apply to your alternative test.

Line 86: My impression was that Penney et al. 2012 was on hoverflies, while the other two are general reviews. When you refer to "this system" I assume you refer to the cuckoo-finch and prinia "system." If you mean another system, such as chase-away models, then I suggest improving the clarity.

Figure 1B. How do you know that the greater complexity of the current (which is very unclear here) is not due to the fact that subtle features are still detectable on the "fresh" eggs but faded away on the old eggs?

Figure 1A. This is quite unclear, from the images it would not seem that the morphs chosen are more complex (assuming we read this left to right on each row). Further, the images are unconvincing to illustrate increasing complexity (i.e., no matter how carefully you choose candidate images it will appear as though you "cherry-picked" particular images) and it isn't clear to the reader what aspects result in greater "complexity" from the main text.

Methods 1: In this case, you used the green channel which we assume approximates the double cone sensitivity; however, your patterns are not purely achromatic. They are chromatic too? Why didn't you use the standard approaches to convert to grayscale, which weights each of the three channels?

Methods 2: Your linear model (Complexity \sim Species + Year + Species:Year) tests whether complexity differs by species or year, controlling for a potential interaction. While the analysis is reasonable (though other constructions may be equally reasonable), it isn't quite clear to me how this relates to your main hypothesis. Does chase-away predict higher or lower complexity in the host vs parasite? I suspect it would but this wasn't clear. Does chase-away predict higher or lower complexity in earlier or later time periods? In this case, you found a slightly significant linear increase in complexity over the years but how would one interpret this? Is the null that there was no linear increase? How does that relate to your hypothesis, which would seem to accommodate repeated non-linear shifts in complexity

6(rather than a linear increase or decrease).

It seems to me that a more natural test would be whether complexity in the host and parasite populations is associated over time. Models used for mutualism, parasitism, and predator-prey dynamics come to mind. The chase-away model predicts that selection from the parasite will confer changes on the host, and (in this case) both track one another such that the fidelity of mimicry is similar yet the complexity of both populations is in flux. Instead, it would make sense to see whether changes in complexity in the host result in corresponding changes in complexity of the parasite that track over time. A quick look at your supplemental figure seems to suggest that this might be the case. You can still demonstrate that there is no statistical difference in the fidelity of mimicry, but this approach would more appropriately consider that complexity is in flux (in both populations) rather than assuming a linear increase or decrease with time.

Image: [unable to attach via the system] I've layered these so you can see how well they track. I would suggest host and parasite boxes to be side by side for each year, or just track their differences. The mutualism, parasitism, and predator-prey dynamics literature have other plotting options (e.g., predator/parasite-prey graphs from Lotka-Volterra models, those comparing two populations without time, come to mind).

Assumptions 1 (related to Methods 2 comment): Considering my comment "methods 2," one potential issue is that the only parasite eggs that are found are those that are well matched (i.e., to a subset of the host population at any particular time). This will exacerbate the issues with heteroscedasticity and may suggest that parasites will "track" host populations by definition (because mismatched eggs are not found/measured). This may impact your current analysis and my suggested analysis. Similarly, are all parasite eggs in historic clutches correctly identified as the parasite's egg?

Assumption 2 (related to Methods 2 comment): The levels of complexity (either higher or lower) that yield significantly greater fitness will differ over time, sometimes less complexity would help differentiating host and parasite eggs while in other years more complexity will help egg recognition.

Data: You have two columns with complexity data, one labelled "a" and one "ac". They are perfectly matched except for the eggs that have "pre" in their names. It is unclear why one is used over the other in the codes. It also isn't clear how you have years with obligate brood parasite eggs but not without their host...

Reviewer #3 (Remarks to the Author):

Comments on MS19066

Title: Rapid evolution of a brood parasite's egg pattern does not lead to large increases in mimetic fidelity

7By quantifying pattern complexity of 414 tawny-flanked prinia (*Prinia subflava*) and 162 cuckoo finch (*Anomalospiza imberbis*) eggs from 1970–2020, this study showed that the parasite, cuckoo finch eggs evolved towards their hosts, the tawny-flanked prinia, and host eggs evolved away from parasites at a similar rate, suggesting that the mimic evolved towards the model, and the model has also evolved away from the mimic.

However, there was no detectible increase in parasitic mimetic fidelity to hosts, supporting the hypothesis that the persistence of imperfect mimicry can be explained by chase-away evolution in models.

In my opinion, this study provided a rare case for the persistence of imperfect mimicry in nature. I enjoy reading this paper and think it was well written.

Therefore, I have only minor comments.

1. They showed that in all analyses, one egg per photographed clutch was included (prinia, $n = 414$; cuckoo finch, $n = 162$).

In another paper (Stevens, Troscianko and Spottiswoode, 2013, *Nat Commun*) showed that 1) the tawny-flanked prinia (*Prinia subflava*) has strong egg rejection (in particular for bad-mimetic eggs), and 2) repeated parasitism by the same cuckoo finch (*Anomalospiza imberbis*) is common in host nests (as an adaptation to increase the probability of host acceptance).

In the case of repeated parasitism by the same cuckoo finch, how did you choose eggs of the cuckoo finch for the 162 nests/eggs?

2. They showed that host evolution can counteract parasite evolution, resulting in the persistence of imperfect mimicry.

Why this occurred? They should discuss a bit in the Discussion.

One possibility is that the tawny-flanked prinia could have cognitive and sensory limitations for egg recognition and egg rejection, thus make a “relaxed selection” for the cuckoo finch, something like that if the host accepts eggs, it is not necessary for the parasite to lay a mimetic egg.

Decision Letter, first revision:

27th January 2023

Dear Tanmay,

Thank you for your letter asking us to reconsider our decision on your Brief Communication entitled "Rapid evolution of a brood parasite's egg pattern does not lead to large increases in mimetic fidelity.". After careful consideration we have decided that we would be willing to consider a revised version of your manuscript.

Along with your revised manuscript, you should also submit a separate point-by-point response to all of the concerns raised by the reviewers, in each case describing what changes have been made to the manuscript or, alternatively, if no action has been taken, providing a compelling argument for why that is the case. If we feel that a substantial attempt has been made to address the reviewers' comments, this response will be sent back to the reviewers - along with the revised manuscript - so that they can judge whether their concerns have been addressed satisfactorily or otherwise.

I should stress, however, that we would be reluctant to trouble our reviewers again unless we thought that their comments had been addressed in full.

- ensure it complies with our format requirements for Articles as set out in our guide to authors at www.nature.com/natecolevol/authors/index.html

- state in a cover note the length of the text, methods and legends; the number of references and the number of display items.

Please ensure that all correspondence is marked with your Nature Ecology & Evolution reference number in the subject line.

Please use the following link to submit your revised manuscript:

[REDACTED]

I would appreciate it if you could tell me if you think you will be able to submit a revised manuscript, and also the likely timescale.

I look forward to hearing from you soon.

[REDACTED]

Author Rebuttal, first revision:

9Reviewers' Comments:

Reviewer #1 (Remarks to the Author):

- There is already a prior paper on this topic, that spans a 100 year long period, and shows that host-parasite mimicry co-evolved towards better match over the time: Geltsch et al. 2017 BJLS. You'll need to please consider in what way your result is novel and not incremental or specific to your study system.

Thank you very much for drawing our attention to this paper.

The study by Geltsch et al indeed studies four measures of egg pattern across time in a brood parasite-host system. However, our study comes to the opposite conclusion. While Geltsch et al. showed that mimicry improved (in one of the four traits they studied), we show that mimicry does not improve. Our finding that coevolution prevents increases in mimetic fidelity is therefore a novel finding in relation to that of Geltsch et al.

A second important distinction between our study and that of Geltsch et al. is that we studied a trait that is used by hosts in rejection decisions, and is therefore likely under selection due to parasitism in both hosts and parasites (Dixit et al. 2022 *Proc R Soc B*). The traits quantified by Geltsch et al are not known to be used by the hosts they studied (Šulc et al. 2019 *Anim Behav* – “as we [do] not know which of the above traits [referring to the same traits that Geltsch et al studied] are the most important cues for host recognition”), and therefore it seems unlikely that temporal changes in these traits in hosts and parasites can plausibly be attributed to selection from brood parasitism.

In summary, our result is novel in that it is the first demonstration that chase-away evolution maintains imperfect mimicry despite rapid evolution of mimics. To highlight this, we have changed our title to “Chase-away evolution maintains imperfect mimicry despite rapid evolution of mimics”.

- I am also wondering about color--color plays a critical role in your study systems' mimicry but it is not considered in my judgement here fully in this paper, even though the senior author has already published on the pattern (and some lack of) coevolution in color between this host and parasite in the early 2010s.

Indeed, colour plays an important role in this system. However, we focussed on complexity as a single trait, which allowed us to make clear predictions about the direction of evolution and test these predictions. We did not seek to study all aspects of colour and pattern mimicry in this study, see e.g.

10Stoddard et al. (2014 *Nat Comms*) for other studies focussing solely on pattern. Also, in this system colour and pattern are uncorrelated (see e.g. Caves et al., 2015, 2017, 2021, all *Proc R Soc B*) and can therefore be studied independently.

- I would like to see Fig. 1b represent the linear analyses, not the historical vs. modern comparisons, since the methods differed between the collections of these eggs/samples and we know that collection specimens fade a lot over years/decades compared to fresh eggs from the field. How did you correct for that factor or is the trait set that goes into your complexity metric not affected by storage over time?

Thank you for this suggestion. A figure showing year-by-year patterns is already presented in extended data figure 1. Although collection methods differed slightly, there was no bias towards photographing or collecting more or less complex egg phenotypes at any time point (see Spottiswoode and Stevens 2012 *Am Nat* for details of lack of bias in the historical collection, as well as for evidence that eggs of our study species do not differ measurably in phenotype upon being blown). Although specimens may fade over time particularly if poorly stored (which was not the case for this collection), this would not affect the trait set that goes into the complexity metric, since no traits relating to colour are extracted. (Also see our detailed response to a similar comment made by Reviewer 2.)

While you have (reasonably, in our opinion) suggested highlighting the linear analyses, reviewer 2 (Dr Daniel Hanley) suggested that instead we highlight the historical vs modern comparisons, because there were two main periods of data collection. We find both alternative perspectives reasonable, and therefore we have retained the original Figure 1b as it is a clearer representation of change over time. This is because it removes the noise present in Extended Data Figure 1, which itself is due to small sample sizes in some years. Nevertheless, because we agree with you that it is good to highlight the linear analysis, we discuss this analysis in detail in the main text, leaving the historical vs modern comparisons to the Methods section.

Reviewer #2 (Remarks to the Author):

- Dear Tanmay et al.,

11I greatly enjoyed your article “Rapid evolution of a brood 1 parasite’s egg pattern does not lead to large increases in mimetic fidelity.” In this article, you explored pattern complexity in a population of an avian brood parasite and its host using a 50-year long-term dataset. Unlike many, you did not assume that the hosts’ eggshell phenotypes were static. Instead, you considered whether reciprocal selection pressures on the parasite and host populations may alter eggshell phenotypes in both, and if so, how this might alter the fidelity of eggshell mimicry. You found that the tawny-flanked prinias’ (hereafter, prinia) eggs were more complex than those of cuckoo finches, that eggshell complexity increased (slightly) over time, and that the level of mimetic fidelity appeared consistent over time.

While I do have suggestions for alternative approaches, it seems the data and the patterns they illustrate are clear enough. In my opinion, the strength of this article is that it considers both host and parasite eggshell phenotypes. In most cases, coevolutionary arms races are assumed and only the egg traits of the parasites and the egg recognition abilities of the hosts are considered. Articles such as this, will help break us from that mold and yield much more insightful arguments about host-parasite dynamics. I hope you find my comments useful in revising your manuscript.

With best wishes,

Daniel Hanley

Dear Daniel, many thanks for your positive comments about this study and your constructive criticism. We greatly appreciate your thorough review.

- Major comments:

You have initially framed your paper around the premise that models evolve away from mimics, and host (rapid) evolution can counteract the eggshell adaptations of their parasites and yield imperfect mimicry. My first three comments will relate to these points, but more general comments will follow.

1. Models evolving away from mimics: In this manuscript you measured eggshell complexity in both hosts and parasites. You found that hosts were more complex than parasites and there was a slight increase in complexity over time, but there was no significant interaction between

12species (host or parasite) and time. First, a particular level of complexity does not imply the same actual pattern on the host and the parasite. Second, I didn't recall seeing a test that showed parasites evolving "toward" hosts or hosts "away" from parasites (see below). Had the interaction been significant, such that complexity of the parasites' eggs were evolving to be more similar to that of the host, then perhaps these statements would be justified. While I understand that you are presented evidence that is consistent with this assumption (e.g., Line 66), it seemed conspicuously odd that you didn't actually test whether hosts were evolving 'away' from mimics directly. Please see my suggestions for other approaches below.

As you say, a particular level of complexity does not imply the same actual pattern. However, this is true for all measures of pattern. For instance, measures of marking size are commonly used in brood parasitism literature. Two patterns can have similar 'marking sizes' yet be visually very dissimilar. The key point is that the metric of complexity we use here does predict egg rejection in this system (Dixit et al. 2022 *Proc. R. Soc. B.*) so it does capture aspects of pattern used in egg discrimination by host sensory and cognitive systems (and so under selection for mimicry), whatever specific aspects of pattern those may be.

Regarding your second point, showing an increase in complexity across species does show that hosts evolve 'away' from parasites, since host eggs are more complex than parasitic eggs. Furthermore, the finding that mimetic fidelity remained constant despite complexity changing over time also highlights that hosts evolved away from parasites. However, if you mean that we did not test for increases in complexity separately in hosts or parasites, then indeed that is correct. The reason for this is that if we did, we would be comparing p-values between these two tests – see misinterpretation 16 in Greenland et al. (2016 *Europ J Epidem*) for why this is not advisable (or for other examples, see <https://elifesciences.org/articles/48175> and <https://www.nature.com/articles/nn.2886>). Second, due to greater variation in host eggs than parasitic eggs, one would require a much larger sample size of host eggs than parasitic eggs to detect the same effect. Therefore, we pooled all eggs into one model to avoid these statistical issues.

- 2. Rapid evolution of parasites: You referred to rapid evolution repeatedly (title and lines 26, 28, 69, and 88); however, I saw no evidence of this. Your work focused on trait complexity, rather than particular colors and patterns. The changes in complexity over time were modest (and not particularly well described linearly – see suggestions below), and you found no appreciable difference in the fidelity of mimicry. So, as the result, I struggled to see what rapid evolution you were talking about. I do understand that in some of these cases you were speaking more generally, that rapid evolution in one population could counteract the evolution in another; however, it did seem that you were stating the parasites were rapidly evolving and their hosts

13were managing to 'keep their distance.' I might remove 'rapid' from the title and include something instead about the evolutionary process itself (e.g., something with "chase").

Our point with 'rapid evolution' was that we found rapid changes in the complexity of eggs. It is true that these changes were modest, but we think these (modest) changes are rapid given that the data spanned 50 years, and our species are vertebrates. We have clarified throughout that when we refer to rapid evolution, we mean rapid changes in egg pattern complexity (not in mimetic fidelity, which, as you point out, has not changed over time). For example, in Lines 98-99 we state that "In summary, tracking model and mimic phenotypic evolution over 50 years showed that despite rapid evolution of parasites)".

We agree that it would be worthwhile changing the title to emphasise the evolutionary process as you suggest. We have changed it to "Chase-away evolution maintains imperfect mimicry despite rapid evolution of mimics".

- 3. Imperfect mimicry: An apparent main conclusion is that chase-away evolution can explain the maintenance of imperfect mimicry. You raised this as an important point in the abstract (line 28) and reintroduced the idea later in the main text (~line 86). While this may be true, I think this point should take a less prominent role. I think that this model is much more powerful as an "alternative" process, and that more emphasis is needed on the evolutionary processes (rather than the consequence). For example, a chase-away model does not require constant fidelity of mimicry, as mimicry can improve and worsen over time via this process (see suggestions below), what is most important is the processes and dynamic between the host and parasite. I was expecting more emphasis on alternative processes (e.g., red queen dynamics) and less on the particular (null) pattern. To be fair, your article was quite short, so I understand if you think it isn't quite fair for me to say that this took a "prominent role." I also realize that the fidelity of mimicry illustrates the evolutionary process, which I'm arguing is your main point. However, because the paper was so short, it also had a few easily digestible take-home messages. For me, this was one (important) one, and will be a useful vehicle for arguing chase-away evolution (since the hosts ability to stay differentiated is evidence of the process). I think it may be useful to consider a slight shift in emphasis. Again, apologies for subtlety here, but it seems that the constant fidelity of mimicry (currently over-emphasized) is evidence of the chase-away evolution (currently under-emphasized).

Thank you for these thoughtful points. We agree that the constant fidelity of mimicry is evidence for chase-away evolution. However, even if mimetic fidelity had not remained constant, this would not imply chase-away evolution had not occurred (as you say – "a chase-away model does not require constant fidelity of mimicry"). Therefore, we discussed the lack of change in mimetic fidelity as a

consequence of chase-away (c.f. Mclean et al 2019 *Q Rev Biol*; Sherratt and Peet-Paré 2017 *Phil Trans R Soc B*). However, we agree that we should have better emphasised chase-away evolution (see our edits in response to point 2).

- As a secondary point relating to “imperfect mimicry,” I think it would be beneficial to make it clearer that the complexity of the eggshell patterns are not perfectly matched. Your study species has nearly perfect eggshell color mimicry in several distinct morphs. While individual host females tend to lay eggs of these similar colors, each has distinct eggshell patterns. Thus, while a cuckoo-finch female may be able to match the colors/pigments for a broad subset of the host population, she will not be able to successfully produce a pattern that matches those same females. Therefore, for those specific features (eggshell patterns) we expect imperfect mimicry. Those patterns would be selected by multiple host females (with differing features) and likely generate some intermediate phenotype that is a reasonable facsimile to those found in the host population. These concepts are only quickly introduced (lines 44-47), but I suspect that the importance of these lines will be lost on most readers.

Thank you for this suggestion. We have emphasised this point that each female lays her own specific eggshell pattern, and thus an individual cuckoo finch cannot match all (or even most) host phenotypes. See Line 43-45 “Individual prinias lay eggs with distinct colour and pattern phenotypes (“egg signatures”; Figure 1A), such that a given cuckoo finch egg will be a poor match to most prinia clutches in the population”. This applies for both colour/pigments and pattern, and indeed parasitised clutches where the cuckoo finch egg(s) is/are poorly-matched to the host eggs are common. As we understand it, your main suggestion here is that we emphasise that cuckoo finches cannot match all host patterns all of the time, which we have done so in Line 43-45, quoted above.

- 4. Analysis: Your data are not particularly continuous. Instead, you have data from two main time periods. Over this time series, you (naturally) have more data on host eggs, which also (naturally) have more variability. You have provided several analyses to overcome the challenges these data might impose, and I think that all were interesting, informative, and it would appear well executed. However, I do struggle to understand why complexity would be linearly related to “year” or how that might relate to your underlying hypothesis. Instead, I think that you actually should analyze how well the parasite population “tracks” your host population over time. This would still allow you to demonstrate the fidelity is constant because both the host and parasite population vary complexity over time. The conclusions that complexity is greater in more recent years is challenging to accept when looking at the data, as it seems that

there are increases and decreases (as one would expect with chase-away evolution) over time. Moreover, host and parasite population seem to track one another. From this perspective a temporal lag, would actually suggest that hosts are evolving away from mimics (or mimics are evolving toward models). These models do not need to be overly complex, there is a rich tradition in population ecology for models that compare two populations over time.

Many thanks for this suggestion. We did not test for “tracking over time” because this is not what the chase-away hypothesis predicts in this system. Given that cuckoo finches are less complex than prinias, selection should favour increased complexity in both species rather than tracking over time – unless there are time periods when cuckoo finches are more complex than prinias. This was not the case (see extended data figure 1).

We have clarified this point in the revision in lines 46-52: “Egg rejection therefore has fitness consequences for both hosts and parasites, and this implies that selection should favour parasites evolving towards hosts (i.e. evolving increased complexity) and hosts evolving away from parasites (i.e. also evolving increased complexity). By quantifying pattern complexity of 414 prinia and 162 cuckoo finch eggs from 1970–2020 (Methods), we tested whether such selection has acted on hosts and parasites in the recent past, and whether this selection led to any change in mimetic fidelity over time.”

- 5. Alternatives: I would appreciate more alternative explanations. For example, red queen dynamics seem like a feasible explanation of the observed patterns. Are there functional or practical differences between these hypotheses? Could they both apply but have slightly different foci? Moreover, if you examine the differences between these populations over time (see the previous suggestions) would seasonal differences in rainfall or diet easily explain the apparent increases and decreases in complexity? Are eggs more complex when they have more surface pigments? Aviles et al. 2017 did find season changes in eggshell pigmentation that related to rainfall, and (at least visually) increases and decreases of complexity appear to also track with ENSO patterns (fluctuations in El Niño and La Niña events). For the record, I’m not asking you to write a paper about climate and eggshell colors; instead, I present this as an example of an alternative that might explain why both populations shift in their eggshell features consistently over time. Overall, I would appreciate it if you could give the alternatives greater attention.

Thank you for this suggestion. We did not emphasise the red queen hypothesis because we did not (and could not) directly measure fitness, and thus test the prediction that relative fitness has remained constant. Yet if we assume fitness is directly related to mimetic fidelity in host and parasite, then red-

queen dynamics are functionally equivalent to the chase-away model we describe here, in that mimetic fidelity has remained constant. Indeed, as emphasised by Brockhurst et al. (2015; *Proc R Soc B*), arms race dynamics and red queen dynamics are different extremes of the same phenomenon. Thus, the red-queen hypothesis is not an alternative explanation.

We agree with your general comment about alternative hypotheses, and have given them greater attention as you suggest. We think it unlikely that selection pressures such as predation or climate would select for differences in complexity; as you rightly say, two patterns could have the same complexity yet be visually very dissimilar (to predators, for instance, for which we do not have behavioural data comparable to our experimental data on host discrimination). Furthermore, the main predators of eggs at our field site are snakes (Spottiswoode and Stevens 2012 *Am Nat*), which rely mostly on olfaction and infra-red. *Prinia* nests are enclosed, meaning that the eggs are generally not visible unless the nest itself is located. We would also not expect climate to select for changes in complexity, both because it is likely that selection from temperature would act more on background colour than pattern complexity (see e.g. <https://www.biorxiv.org/content/10.1101/559435v2.abstract>; Lahti & Ardia 2016 *Am Nat*), and because any selection from increased temperatures would likely select for reduced complexity over time.

We have discussed such alternatives in Lines 88-97: “Although observed changes in complexity conformed to *a priori* predictions of coevolution, this study is correlational. We must therefore consider alternative explanations which could influence host and parasitic eggs in tandem, such as selection on egg pattern complexity from predation or climate. However, the main predators at our field site are snakes, which rely mostly on olfaction and infra-red, and *Prinia* nests are enclosed, limiting egg visibility at long range⁸. Climate change also appears unlikely to select for increases in complexity, since increased temperatures are likely to select for fewer pattern markings (which absorb more heat than unmarked eggshells)⁹. Complexity is highly correlated with the number of pattern markings and weakly correlated with pattern coverage⁷; thus, increased ambient temperatures due to climate change should select for reduced complexity, contrary to our findings.”

- Minor comments:

Line 43: It is great that you have long-term data, and these data should be promoted. However, in reality, your dataset has two periods of active collection. While this has some statistical implications, I think this suggests you should exhibit some caution on such claims.

Thank you for this suggestion. Although reviewer 1 felt that we should emphasise the linear analysis, we take your point that with two main periods of data collection, we should focus on changes in complexity between the historical and present-day datasets. We found both perspectives valuable, and therefore we illustrate the two periods in Figure 1b, but focus on the linear analysis mostly in the text. Please see our response to reviewer 1 for details.

- Line 50: This is a truly fantastic time series. I would presume these are relatively evenly distributed through time? For example, ~8 host eggs each year (but I'm not asking for these details here). Upon closer inspection, it is clear that you have good sampling in only a few of these years.

Indeed, see our comment above. Incidentally, this also means that we cannot test whether populations track each other over time (fluctuations are highly likely to be due to noise) – but in any case, the chase-away hypothesis does not predict tracking over time in this system (see response to your comments about this below).

- Line 54: It isn't clear to me how you are presented complexity (log scale) as a percentage. Perhaps another approach would be to keep the metrics in their units but discuss their effect sizes as odd ratios or similar metrics.

As we did not conduct any analysis of logits, we could not use odds ratios. Given that effect sizes on logarithmic scales are unintuitive, we used percentages, which have a natural mathematical relationship to logarithms. In the methods we state that we expressed complexity changes as a percentage by calculating $\exp(\text{Estimate})$ (lines 167-169).

- Line 56: Why would complexity have increased significantly over the last 50 years unless you caught the hosts and parasites early in their interaction? A chase-away model would imply that complexity -changes- not necessarily gets more complex. A closer inspection of your supplemental figure shows that this is the case, the complexity fluctuates in both populations repeatedly throughout the time series. This calls into question any expected linear differences between complexity and time. In fact, it isn't clear why this is an expected relationship under the chase-away model at all (when the hypothesis simply states that the model should evade the mimic). This requires more explanation.

The reason that complexity should change over time is that hosts are more complex than priniads. Therefore, selection should favour parasites to evolve higher complexity, and hosts to also evolve higher complexity. Were parasites to catch up to hosts, then we would expect tracking. But at present, a temporary decrease in host complexity in one year should not select for reduced complexity in cuckoo finches, since cuckoo finch eggs are still less complex than prinia eggs. Regarding the fluctuation in the supplemental figure, this seems largely likely to be an artefact of small sample sizes in some years.

We have clarified this point in the revision in lines 46-52: “Egg rejection therefore has fitness consequences for both hosts and parasites, and this implies selection should favour parasites evolving towards hosts (i.e. evolving increased complexity) and hosts evolving away from parasites (i.e. also evolving increased complexity). By quantifying pattern complexity of 414 prinia and 162 cuckoo finch eggs from 1970–2020 (Methods), we tested whether host and parasitic phenotypes have changed in the predicted direction in the recent past, and whether such reciprocal evolution led to any change in mimetic fidelity over time.”

- Line 56: I would prefer for these data to remain in the log scale with the appropriate CI. These details will be in your methods, and are useful for those reading the statistics within the parentheses. You could/should present these as odd ratios for a more intuitive way to present the information to all readers.

Please see our comment to line 54. While expressing effect sizes as percentages is unusual, effect sizes in the log scale give the reader no sense of the size of the effect. We could not present effect sizes as odds ratios (see comment to line 54).

- Line 61: Is the greater variance in the host population actually a feature of the chase-away model, or simply a statistical (/practical) artefact?

Good question! It seems likely that it is a feature of hosts having diversified as a result of selection from parasitism. Indeed, if we (repeatedly) re-sample the host dataset down to the sample size of the parasite dataset and calculate the variance, we find that the average variance in complexity of hosts (1,171,997 \pm 178,654 SD is still approximately 9 times as much as that of parasites (133,586; no SD as there is no repeated resampling here), and we obtain similar results when we take the natural logarithm of complexity. This indicates that hosts really do exhibit higher diversity than parasites, and that the higher variance in hosts is not simply due to higher sample size. We briefly mention this point in the main text (Lines 62-63: “hosts exhibiting higher variance in complexity than parasites, likely as a result of

diversifying selection on host phenotypes”), though we do not present the back-of-the-envelope calculation above, in order to retain focus on the main message of the paper.

- Line 66: It may be my preference, but I do not think that the 1 and 2 are really necessary. The sentence and take-away message are simple enough without them.

Thank you for this suggestion. We have removed them.

- Line 68: In my opinion, you may benefit from merging the content from current lines 68-75 on line 60 (just before the new sentence). You could likely condense your alternative test explanations (62-64) and then focus more on interpretation, which felt a little lacking and lines 65-67 felt premature. They would have much more weight after your full presentation and strengthen your last paragraph or set it up, the last sentence of your penultimate sentence. If you choose to follow this advice, you may need to rework some of the text a little.

Thank you for this suggestion. While we understand your reasons for suggesting this restructuring, we found that presenting all results together (as you suggest) might be confusing for readers, since the finding that mimetic fidelity remained constant makes sense only after the interpretation that there was no difference between the rate of evolution in parasites and hosts. To emphasise this, we have used an if-then statement in lines 69-71: “If chase-away evolution in hosts occurred at a similar rate to parasite evolution, as implied above, then we would expect to see limited increases in mimetic fidelity despite rapid evolution of parasites.”

We also agree that lines 65-67 could feel premature. We have slightly edited these lines to emphasise how the conclusion follows from the results in Lines 66-68: “Overall, the finding that complexity increases over time suggests that parasites have evolved towards hosts, and that hosts have evolved away from parasites at a similar rate”.

- Lines 73-75: You are swapping between log scales and percentages here. Also, the difference between the two log scales is 0.04. However, unless I'm missing something (which is entirely possible) a 0.04 difference is not the same thing as a 4% increase. Finally, you state that there is “no significant” increase, but the overlap is greater for your historical data. So, shouldn't that be “This [does not represent a] [...] significant [decrease] in this trait-based measure of...” I've added brackets for added, removed, or altered words and I think that this section could be more carefully worded.

20Additionally, I find this a very confusing presentation. Instead, I strongly recommend illustrating the two bootstrapped distributions. It would be much more intuitive to show those and describe the overlap.

Thank you for this suggestion. Given that percentage differences are calculated as $\exp(\text{Estimate})$, due to the natural relationship between percentages and logarithms, it is true that a 0.04 difference equals a 4% difference. We believe that the use of the word 'increase' is justified because even though the overlap is greater for historical data, we are testing for an increase in mimetic fidelity (i.e. decrease in host-parasite difference). Finding no increase in mimetic fidelity (despite rapid evolution in parasites) is a key message of the paper, as highlighted by our revised title.

We understand, however, the confusion with swapping between log scales and percentages. Unfortunately, we cannot use percentages to describe complexity differences in the historical and current dataset, only to describe changes over time in mimetic fidelity. To only use percentages, therefore, we would have to remove the sentence "Bootstrapped estimates of historical and current mimetic fidelity showed considerable overlap (mean historical complexity difference on a logarithmic scale = 0.46, 95% CI = {0.38,0.55}; mean current complexity difference = 0.42, 95% CI = {0.39,0.45}; Figure 1C)." We are happy to do this if you and/or the editor think it would be beneficial, but for now we have not removed it, because we think it is useful to readers to highlight the two distributions and the narrow confidence intervals.

Thank you also for your suggestion to illustrate the two distributions (hosts and parasites) - see a rough version below. However, we think this is less clear, as an illustration that mimetic fidelity hasn't changed, than Figure 1c – the reader must compare historical cuckoo finches with prinias (left panel), compare current cuckoo finches with prinias (right panel), and then compare the two comparisons (i.e. compare the extent of overlap in left and right panels)! By contrast, Figure 1c shows at once that mimetic fidelity does not change over time.

Also, the distributions of hosts and parasites are already shown in the paper as Figure 1b. We bootstrapped the host-parasite differences, not the host and parasite datasets individually (see methods, lines 221-230), and thus the image below is identical to Figure 1b except that the boxes are in a different order. Therefore, we propose to retain Figure 1c, which shows that mimetic fidelity (y axis) hasn't changed over time (x axis).

- Line 73: This is a stylistic preference (ignore as you see fit - or follow the journal's suggestions). For your confidence intervals I like to present these as 0.38 to 0.55. Presenting them like this would avoid the curly brackets which are awkward, especially when alongside the parentheses.

Thank you for this suggestion. We have changed our presentation to the format “95%CI = -0.9–0.1%”

- Line 75: I suggest a slightly different presentation here. I would start by describing the overlap in mimetic fidelity and then the performance of your discriminant analysis to differentiate parasite eggs from host eggs. Presenting these as two complementary tests, rather than a primary and follow-up test would be useful here.

Thank you for this excellent suggestion. We have done this by stating “To quantify changes in mimetic fidelity, we calculated all host-parasite complexity differences from 1970–2002 (historical) and from 2012–2020 (current) (Methods). Bootstrapped estimates of historical and current mimetic fidelity showed considerable overlap (mean historical complexity difference on a logarithmic scale = 0.46, 95% CI = 0.38–0.55; mean current complexity difference = 0.42, 95% CI = 0.39–0.45; Figure 1C). This corresponds to no significant increase in this trait-based measure of mimetic fidelity (bootstrapped estimated increase = 4%, 95% CI = -3%–12%, Figure 1C). We also independently estimated mimetic

fidelity using a discriminant analysis based on complexity..." (Lines 71-78).

- Line 76: I appreciate that you are providing an alternative approach (and I like the approach you used); however, do you have any reason to assume this is a false negative? The reasons would be useful here. Depending on your reason, it may also apply to your alternative test.

We had no reason to assume this was a false negative, and simply wished to provide an alternative, equally valid approach. Therefore, we have removed any discussion of false negatives (see our response to your comment for Line 75 for the new text). Thank you for pointing out that our phrasing here gave the wrong impression!

- Line 86: My impression was that Penney et al. 2012 was on hoverflies, while the other two are general reviews. When you refer to "this system" I assume you refer to the cuckoo-finch and prinia "system." If you mean another system, such as chase-away models, then I suggest improving the clarity.

Thank you – we have stated that we are referring to “this host-parasite system”. Indeed, Penney et al. is not a general review but rather a comparative analysis. We have removed this citation when referring to hypotheses for imperfect mimicry, such that we only cite reviews of the topic.

- Figure 1B. How do you know that the greater complexity of the current (which is very unclear here) is not due to the fact that subtle features are still detectable on the "fresh" eggs but faded away on the old eggs?

Thank you for this thought. As we understand, it is largely blue-green colours on eggs which fade (e.g. Cassey et al., 2010 *Behav Ecol Sociobiol*), which has no relevance to the pattern measures we extracted, since pattern measures extracted from NaturePatternMatch should be unaffected by the underlying colour. Old eggs were photographed in 2007 and 2009, and as discussed in Spottiswoode and Stevens (2012, *Am Nat*), eggs were kept in a darkened room and collected relatively recently. Furthermore, subjectively NaturePatternMatch seems no better or worse at detecting features on old vs new eggs. Therefore we do not expect any fading to influence complexity.

It is worth noting that in the event that fading affected the detectability of faint markings – though this is unlikely, see above – any background colour fading on old eggs would make faint markings *more* detectable on these eggs, resulting in higher complexity scores for old eggs than fresh eggs. Our results run counter to this, and are therefore conservative.

We now include the details above in Lines 170-179 of the Methods section: “One concern with using historical egg collections is that the background colour of eggs can fade over time, especially if they are poorly stored, which was not the case for the eggs photographed as part of this study. Old eggs were photographed in 2007 and 2009, and eggs were kept in a darkened room and collected relatively recently⁸. Furthermore, it is largely blue-green colours on eggs which fade (e.g. ¹⁵), which has no relevance to the pattern measures we extracted, since pattern measures extracted from NPM should be unaffected by the underlying colour. In the unlikely event that fading affected the detectability of faint markings by NPM, any background colour fading on old eggs would make faint markings *more* detectable on these eggs, resulting in higher complexity scores for old eggs than fresh eggs. Our results run counter to this (see Main Text), and are therefore conservative.”

- Figure 1A. This is quite unclear, from the images it would not seem that the morphs chosen are more complex (assuming we read this left to right on each row). Further, the images are unconvincing to illustrate increasing complexity (i.e., no matter how carefully you choose candidate images it will appear as though you "cherry-picked" particular images) and it isn't clear to the reader what aspects result in greater "complexity" from the main text.

Thanks for this suggestion. We have removed the words “illustrating changes in complexity over time” from the figure caption, and simply said that they are “from the historical (left) and current-day (right) samples”; these images were chosen randomly. We do wish to illustrate the diversity in phenotypes across time points for transparency and to show what sort of complex phenotypes we are studying, but we agree that these images do not convincingly illustrate increased complexity (as indeed they cannot in such a small sample).

- Methods 1: In this case, you used the green channel which we assume approximates the double cone sensitivity; however, your patterns are not purely achromatic. They are chromatic too? Why didn't you use the standard approaches to convert to grayscale, which weights each of the three channels?

The standard approach in this system has been to convert to greyscale using the green channel (see e.g. Spottiswoode and Stevens 2010, 2011, 2012; Stoddard et al 2019, Caves et al., 2015, 2017, 2019), and also in other systems where the green channel approximates vision (e.g. Howard et al. 2019 *Curr Zool*; Gómez et al. 2021 *Plos ONE*, Stoddard and Stevens 2010 *Proc R Soc B*) though we appreciate your suggestion. Furthermore, the measure of complexity that predicted rejection in Dixit et al. (2022, *Proc R*

Soc B) was based on using the green channel, and therefore to study how this biologically-relevant measure of complexity changes over time, we had to use the same methods.

- **Methods 2:** Your linear model (Complexity ~ Species + Year + Species:Year) tests whether complexity differs by species or year, controlling for a potential interaction. While the analysis is reasonable (though other constructions may be equally reasonable), it isn't quite clear to me how this relates to your main hypothesis. Does chase-away predict higher or lower complexity in the host vs parasite? I suspect it would but this wasn't clear. Does chase-away predict higher or lower complexity in earlier or later time periods? In this case, you found a slightly significant linear increase in complexity over the years but how would one interpret this? Is the null that there was no linear increase? How does that relate to your hypothesis, which would seem to accommodate repeated non-linear shifts in complexity (rather than a linear increase or decrease).

Because hosts are more complex than parasites, parasites should evolve directionally towards hosts (i.e. elevated complexity) and hosts should evolve directionally away from parasites (chase-away, also towards elevated complexity). Thus, this hypothesis does predict directional changes in complexity and so we used a linear model to test for directional changes rather than testing for tracking of phenotypes over time. The null would be no linear change over time.

We have clarified this point in the revision in lines 46-52: "Egg rejection therefore has fitness consequences for both hosts and parasites, and this implies that selection should favour parasites evolving towards hosts (i.e. evolving increased complexity) and hosts evolving away from parasites (i.e. also evolving increased complexity). By quantifying pattern complexity of 414 prinia and 162 cuckoo finch eggs from 1970–2020 (Methods), we tested whether host and parasitic phenotypes have changed in the predicted direction in the recent past, and whether such reciprocal evolution led to any change in mimetic fidelity over time."

- It seems to me that a more natural test would be whether complexity in the host and parasite populations is associated over time. Models used for mutualism, parasitism, and predator-prey dynamics come to mind. The chase-away model predicts that selection from the parasite will confer changes on the host, and (in this case) both track one another such that the fidelity of mimicry is similar yet the complexity of both populations is in flux. Instead, it would make sense to see whether changes in complexity in the host result in corresponding changes in complexity of the parasite that track over time. A quick look at your supplemental figure seems to suggest that this might be the case. You can still demonstrate that there is no statistical difference in the

25fidelity of mimicry, but this approach would more appropriately consider that complexity is in flux (in both populations) rather than assuming a linear increase or decrease with time.

Image: [unable to attach via the system] I've layered these so you can see how well they track. I would suggest host and parasite boxes to be side by side for each year, or just track their differences. The mutualism, parasitism, and predator-prey dynamics literature have other plotting options (e.g., predator/parasite-prey graphs from Lotka-Volterra models, those comparing two populations without time, come time mind).

Thank you for this suggestion; see our comments above, but briefly, we predicted and tested for directional change rather than tracking because parasite complexity is consistently lower than host complexity. Any apparent tracking seems likely due to small sample sizes in specific years – see the rough figure below (Species A = cuckoo finch; Species P = prinia). In years with high sample sizes, we do not observe tracking. We would be happy to replace Extended Data Figure 1 with the figure below if you and/or the editor feel that this would be clearer. However, we have not done this for now, because we think the 'side-by-side' plots in the current version of Extended Data Figure 1 are easier to digest, and more transparently show which years had high and low sample sizes for each species.

- Assumptions 1 (related to Methods 2 comment): Considering my comment “methods 2,” one potential issue is that the only parasite eggs that are found are those that are well matched (i.e., to a subset of the host population at any particular time). This will exacerbate the issues with heteroscedasticity and may suggest that parasites will “track” host populations by definition (becF2007ause mismatched eggs are not found/measured). This may impact your current analysis and my suggested analysis. Similarly, are all parasite eggs in historic clutches correctly identified as the parasite’s egg?

All parasitic eggs in historical clutches were correctly identified; we confirmed this. Identity can be reliably assigned from the presence (hosts) vs absence (parasite) of “scribble” markings (Lines 132-133).

Regarding eggs being found, often hosts take a little time to reject a poorly matched egg (particularly eggs that are poorly matched in terms of pattern, rather than colour), so there was certainly opportunity

for poorly matched eggs to be found (and many were). Furthermore, given the high variation in host eggs, all cuckoo finch eggs are poor matches to the majority of the host population at any given time. We note these points in Methods Lines 180-187: “A second concern with studying host and parasitic egg phenotypes more generally is that some (likely poorly-matched) parasitic eggs may be rejected from host nests before data from that nest are collected. This may mean that only closely-matched parasitic eggs are phenotyped. However, in this system this is unlikely to be a problem, since (i) hosts often take 1–4 days to reject a poorly-matched egg (particularly eggs that are poorly matched in terms of pattern, rather than colour), and (ii) high variation in host egg appearance between clutches (Figure 1B) means that all cuckoo finch eggs are poor matches to the majority of the host population at any given time. Thus, there is unlikely to be a bias towards phenotyping well-matched eggs.”

- Assumption 2 (related to Methods 2 comment): The levels of complexity (either higher or lower) that yield significantly greater fitness will differ over time, sometimes less complexity would help differentiating host and parasite eggs while in other years more complexity will help egg recognition.

This seems unlikely, as overall host eggs were more complex than parasitic eggs. Therefore selection should favour elevated complexity in both species. See our responses above explaining why we expect selection to act in this direction.

- Data: You have two columns with complexity data, one labelled “a” and one “ac”. They are perfectly matched except for the eggs that have “pre” in their names. It is unclear why one is used over the other in the codes. It also isn't clear how you have years with obligate brood parasite eggs but not without their host...

Thank you for spotting this oversight. We have removed the column “ac” since this referred to eggs where multiple images of the phenotype were taken (see Dixit et al. 2022 *Proc R Soc B*), which is irrelevant for this study.

In some years, some host eggs were not photographed or analysed due to a specific research focus on parasitic eggs, and host eggs were not routinely photographed owing to time constraints. We now note this in Lines 130-132 of the Methods: “In a few years, some host eggs were not photographed or analysed due to a specific research focus on parasitic eggs, and host eggs were not routinely photographed owing to time constraints.”

Reviewer #3 (Remarks to the Author):

- Comments on MS19066

Title: Rapid evolution of a brood parasite's egg pattern does not lead to large increases in mimetic fidelity

By quantifying pattern complexity of 414 tawny-flanked prinia (*Prinia subflava*) and 162 cuckoo finch (*Anomalospiza imberbis*) eggs from 1970–2020, this study showed that the parasite, cuckoo finch eggs evolved towards their hosts, the tawny-flanked prinia, and host eggs evolved away from parasites at a similar rate, suggesting that the mimic evolved towards the model, and the model has also evolved away from the mimic.

However, there was no detectible increase in parasitic mimetic fidelity to hosts, supporting the hypothesis that the persistence of imperfect mimicry can be explained by chase-away evolution in models.

In my opinion, this study provided a rare case for the persistence of imperfect mimicry in nature. I enjoy reading this paper and think it was well written.

Many thanks for your positive comments; we are glad you enjoyed the paper!

- Therefore, I have only minor comments.

1. They showed that in all analyses, one egg per photographed clutch was included (prinia, $n = 414$; cuckoo finch, $n = 162$).

In another paper (Stevens, Troscianko and Spottiswoode, 2013, *Nat Commun*) showed that 1) the tawny-flanked prinia (*Prinia subflava*) has strong egg rejection (in particular for bad-mimetic eggs), and 2) repeated parasitism by the same cuckoo finch (*Anomalospiza imberbis*) is common in host nests (as an adaptation to increase the probability of host acceptance).

In the case of repeated parasitism by the same cuckoo finch, how did you choose eggs of the cuckoo finch for the 162 nests/eggs?

29

One parasitic egg was selected randomly in cases of repeated parasitism (see Methods line 122).

- 2. They showed that host evolution can counteract parasite evolution, resulting in the persistence of imperfect mimicry.

Why this occurred? They should discuss a bit in the Discussion.

One possibility is that the tawny-flanked prinia could have cognitive and sensory limitations for egg recognition and egg rejection, thus make a “relaxed selection” for the cuckoo finch, something like that if the host accepts eggs, it is not necessary for the parasite to lay a mimetic egg.

Thank you for this thought. The point here is simply that *because* hosts evolve away from parasites, imperfect mimicry persists. If hosts did not evolve in response to parasites (while parasites evolve towards hosts), we would see an increase in mimetic fidelity over time. Moreover, in a previous study we have shown that there is selection pressure from tawny-flanked prinias with respect to this particular trait, since tawny-flanked prinias reject eggs that differ in complexity from their own (Dixit et al. 2022 *Proc R Soc B*), implying that selection is not relaxed.

We have now emphasised that imperfect mimicry is a result of chase-away evolution in hosts in various places in the manuscript, such as the title, and in Lines 69-71: “If chase-away evolution in hosts occurred at a similar rate to parasite evolution, as implied above, then we would expect to see limited increases in mimetic fidelity despite rapid evolution of parasites.”

Decision Letter, second revision:

30th June 2023

Dear Mr Dixit,

Your manuscript entitled "Chase-away evolution maintains imperfect mimicry despite rapid evolution of mimics." has now been seen by three reviewers, whose comments are attached--I apologise in the delay in getting these reports to you, caused by reviewers being unavailable. The reviewers have

30raised a number of concerns which will need to be addressed before we can offer publication in Nature Ecology & Evolution. We will therefore need to see your responses to the criticisms raised and to some editorial concerns, along with a revised manuscript, before we can reach a final decision regarding publication.

The manuscript has clearly made progress, but reviewer 2 has a number of concerns that still need to be addressed before the manuscript is published

We therefore invite you to revise your manuscript taking into account all reviewer and editor comments. Please highlight all changes in the manuscript text file [OPTIONAL: in Microsoft Word format].

- * Include a "Response to reviewers" document detailing, point-by-point, how you addressed each reviewer comment. If no action was taken to address a point, you must provide a compelling argument. This response will be sent back to the reviewers along with the revised manuscript.
- * If you have not done so already please begin to revise your manuscript so that it conforms to our Brief Communication format instructions at <http://www.nature.com/natecolevol/info/final-submission>. Refer also to any guidelines provided in this letter.
- * Include a revised version of any required reporting checklist. It will be available to referees (and, potentially, statisticians) to aid in their evaluation if the manuscript goes back for peer review. A revised checklist is essential for re-review of the paper.

[REDACTED]

Nature Ecology & Evolution is committed to improving transparency in authorship. As part of our efforts in this direction, we are now requesting that all authors identified as 'corresponding author' on published papers create and link their Open Researcher and Contributor Identifier (ORCID) with their

31account on the Manuscript Tracking System (MTS), prior to acceptance. ORCID helps the scientific community achieve unambiguous attribution of all scholarly contributions. You can create and link your ORCID from the home page of the MTS by clicking on 'Modify my Springer Nature account'. For more information please visit www.springernature.com/orcid.

[REDACTED]

Reviewer expertise:

as before

Reviewers' comments:

Reviewer #1 (Remarks to the Author):

Thank you for addressing my comments in detail, you have done a good job to argue for the novelty of your finding. I still find the analyses wanting with respect to color, however, which is used for mimicry and egg recognition in your system, even if it's independent from the spotting complexity itself.

Reviewer #2 (Remarks to the Author):

Dear Mr. Dixit and colleagues,

Thank you for taking the time to incorporate some of my suggestions and feedback and for providing such detailed responses. I think you did a fine job revising your manuscript. You demonstrate chase-away selection in an avian brood parasite system that displays aggressive mimicry. In this system the model host, the tawny-flanked prinia (prinia), has more complex eggs than their mimics, the parasitic cuckoo finch. You show chase away selection by demonstrating that complexity increases over time for both species, as expected, yet the fidelity in complexity remains constant. While there are some caveats that we have discussed (previously, and a bit more below), you presented an interesting and highly useful case-study for our field. As you rightly point out, little research has focused on selection of both models and their mimics. Thus, your paper will be valuable for broad audiences.

Overall, I think you did a nice job of addressing my concerns, while balancing the other concerns that were raised. My most major critique is that your fine-grained data does appear to reveal more

32intricate, cyclical, patterns. In your defense, you claim that these are driven by sample size, a claim I think would value from a bit more evidence (e.g., a test in a supplement). That said, Figure 1 does show the expected pattern and overall you have presented a complex topic neatly and succinctly.

I hope you haven't found my critiques overly critical. Likewise, I hope the comments were constructive and helpful. At this stage, the paper may require some minor tweaking but I do not feel I can provide any more feedback beyond what we discussed thus far. Good luck with your manuscript.

Please find short responses and thoughts to this version (that I hope provide further assistance).

Best wishes,

Daniel

Comments on the previous edits:

1. Two points

- a. Differences in complexity do not relate to the actual patterns themselves. While I do understand that our field often uses two-tailed tests to analyze trait differences without giving too much thought to the traits themselves, in your case I do think it will pay to mention that prinia pay attention to the complexity (as a trait itself) rather than their paying attention to the complex trait. Perhaps some argument about neural processor, or simply a statement that we simply do not yet know everything they pay attention to. I do see one line early on, but the idea that hosts are responding to an abstract concept of 'complexity' deserves a bit more attention given its importance to your study.
- b. Direction of evolution between hosts and parasites: While I do understand that hosts were always more complex than parasites, they were not always evolving to be ever-more complex, while parasites followed in suite. Your new plot and data show apparent cyclical patterns, with a decrease in complexity in both host and parasite by the mid- to late-1980s, then a period of increase complexity in both by the 2010s, with a decline in parasite complexity by the end of that decade. If this was directional, my expectation would be that the prinia would be more complex, the parasite would get more complex, which would drive more complexity in the prinia, etc. I do see that overall, this is the case (see Figure 1), however your other plot from the response reveals a slightly more nuanced pattern.

2. The evolution didn't seem to be very rapid: As pointed out above there seem to be consistent cyclical changes. Thanks for emphasizing the chase-away mechanism. I will not press you over how quickly evolution must proceed to be considered 'rapid'.

3. Two comments

- a. Comments about imperfect mimicry: Great changes! Thank you.
- b. Comments about how imperfect mimicry works on spots in this system: Also, great work.

4. Expectation of a linear relationship between year and complexity. Given that your data show no particular linear relationship with time (see the plot you inserted into the response letter, which was evident from your last submission), I still am quite perplexed why you expect a linear relationship

33between complexity and time in your dataset. It's clear that hosts are more complex than parasites, but complexity increases and decreases for both in more or less similar fashions. I cannot imagine how complexity would linearly relate to time (also since the dataset really has two time-frames, it may be questionable to do so). Your approach to handle divergent opinions on how to handle the linear analyses vs historical to modern comparisons, seems fair. Overall, your analysis seems OK. However, these data do suggest there's a few other interesting patterns.

5. Alternatives: You have now provided some reasonable alternatives. Your point on red queen is fair enough, and well taken.

Other minor comments:

A. Cyclical patterns in complexity among hosts and parasites rather than linear increases: I am not convinced (but easily could be) that the apparent cyclical patterns between hosts and parasites are a produce of sample size. Was sample size positively related to complexity? I do not disagree with this (true) statement "The reason that complexity should change over time is that hosts are more complex than priniias." And while I do understand that chase away would suggest increasing complexity (linear increases) since the host was originally more complex.... Given the patterns that you have plotted, I do think it's important to explain why they do not become more complex consistently over every sampled year. Instead, they change periodically and symmetrically in their degree of complexity. The reason I mentioned some global and environmental factors, which are known to impact some aspects of egg pigmentation, is that those factors could explain why both hosts and parasites would shift in tandem across these decades, but why hosts may retain their greater complexity. As both species live within the same environment and are subjected to the same pressures. One way or the other, it seems vital to explain why you expect the host and parasite to get more complex when it's clear they simply change synchronously in complexity. If it is true that this is a statistical artifact you would need a test to demonstrate that sample size is related to complexity (so years with lower sample sizes have lower complexity because there are too few eggs). However, unfortunately, you would also need the accompanying test to show that the greater complexity in hosts is not only because you have greater sample size of host eggs (i.e., to show that those observations are not simply because of the sample size – I do not believe this is the case, see my next point).

B. I think the greater variance in the host would actually be a feature of this dynamic, but realize that this is a bit speculative. Great response though.

C. Presentation on original line 75: Thanks for embracing this suggestion. I think the revised passage works nicely!

D. Logs and percentages: Thanks for clarifying. The presentation is still a little challenging, but you did a good job explaining why they are presented in this fashion. I apologize about my misleading comment about 0.04 log units not equaling 4%. Maybe I thought you used log₁₀ (I'm honestly not sure what I was thinking there, I'm sorry); however, while we do get 4%, if 0.04 is assigned to x in this equation, $(\exp(x)-1)*100$, this is increasingly untrue with greater values of x. Hence, my unease in seeing percentages to describe these values. Again, you did a fair job at explaining, but this will likely be a place that is hard to follow in your paper for most readers.

E. Fading: I think your approach is fine. However, my concern is that the fading that occurs will impact how complexity gets quantified from your images as it impacts the relative contrast between spots and background coloration, among other features that relate to capturing and assessing complexity. So, while it may be true that for some morphs you would expect more contrast, this isn't going to be true across the board. There are strong nonlinear relationships with fading specific to each pigment, which occur through oxidation and photooxidation. That said, I think you've done all you can do and added reasonable citations and statements about the limitations.

F. Estimation of the achromatic channel: Totally reasonable to use the green channel. I think the reasoning here is that it most closely matches the peak sensitivity of the double cone. My thinking was that for a standard image you may be better off with a normal grayscale image, which is calculated differently. Your rationale is reasonable and justified.

Reviewer #3 (Remarks to the Author):

Comments on 19066_2

Title: Chase-away evolution maintains imperfect mimicry despite rapid evolution of mimics

I found the authors addressed well with my comments, and thus I have no further comments.

In addition, by reading the authors' response to comments by other reviewers, I found the revision more focused.

Compared with molecular-based studies (e.g., genomes), time-consuming field studies would have few opportunities to get published in high-impact factor journals, and in turn, this would make more and more scientists lose their passion to work in the field.

For example, in recent years, there has not been a good trend that more and more papers were based on other papers' data (even such data was in problem), or just a meta-analysis using other studies' data. This would make research (in particular, ecological research) become a "bad" science.

In light of this, by using over 50 years' data, this study tracked the model and mimic phenotypes and showed that despite rapid evolution of parasites, there was no detectible increase in their mimetic fidelity to hosts, suggesting that the coevolutionary response in hosts was strong enough to prevent increases in mimetic fidelity, and the persistence of imperfect mimicry can be explained by chase-away evolution in models.

*****END*****

35Author Rebuttal, second revision:

Reviewers' comments:

Reviewer #1 (Remarks to the Author):

Thank you for addressing my comments in detail, you have done a good job to argue for the novelty of your finding. I still find the analyses wanting with respect to colour, however, which is used for mimicry and egg recognition in your system, even if it's independent from the spotting complexity itself.

Thank you for looking at this manuscript again; we are glad that you agree that the finding is novel and interesting.

As you say, colour is important in this system. Nevertheless, it would be inappropriate to incorporate colour into this particular study for a few reasons:

1. Colour does not have a “magnitude”, unlike a trait such as complexity. Therefore, in testing for directional changes in complexity over time, we could not incorporate colour in a similar way.
2. The particular question we were looking at here was whether chase-away evolution can prevent increases in mimetic fidelity. We did not seek to incorporate all traits which contribute to mimicry and egg recognition – indeed much of the variation in mimicry and egg recognition remains unexplained by models of rejection (see e.g. Spottiswoode and Stevens 2010 PNAS, Stoddard et al., 2019 Phil Trans R Soc B, Dixit et al., 2022 Proc R Soc B, Dixit et al., 2023 Biol Lett), as is the case in all studies of egg rejection. Given this constraint, we decided to focus on a particular well-defined trait (complexity), for which we could clearly quantify host-parasite differences, and make a priori predictions about the direction of evolution. This is not possible with colour because, as mentioned above, colour does not have a magnitude, and is a multidimensional trait for which it is difficult to quantify overall host-parasite differences.
3. While one host egg colour is not mimicked by parasites (Spottiswoode and Stevens 2012 Am Nat), colour is not a trait which differs, overall, between host and parasitic eggs (Dixit et al., 2023 Biol Lett). In other words, it is not only difficult to quantify mimetic fidelity in colour (for the reasons laid out in points 1 and 2), but colour is not a trait for which selection should drive mimics directionally towards models. Rather, fluctuations in selection are likely, and have been considered in previous work (Spottiswoode and Stevens 2012 Am Nat). By contrast, to test whether chase-away selection prevents increases in mimetic fidelity, one needs to focus on a

trait for which selection should be acting to drive mimics towards models, and models away from mimics. Unlike colour, complexity fits this criterion.

For these reasons, and although we fully agree that colour is an important trait in this system, we believe that our analyses were appropriate in testing the specific evolutionary question we were asking.

We have also added a sentence to the methods section to explain why we focussed on complexity rather than other traits such as colour: “Although many traits may be important for egg rejection in this system⁶, we focussed on complexity because quantifiable differences between hosts and parasites in this trait allow us to make clear predictions about the direction of evolution, namely that both should evolve towards higher complexity⁷.” (Lines 120-123).

Reviewer #2 (Remarks to the Author):

Dear Mr. Dixit and colleagues,

Thank you for taking the time to incorporate some of my suggestions and feedback and for providing such detailed responses. I think you did a fine job revising your manuscript. You demonstrate chase-away selection in an avian brood parasite system that displays aggressive mimicry. In this system the model host, the tawny-flanked prinia (prinia), has more complex eggs than their mimics, the parasitic cuckoo finch. You show chase away selection by demonstrating that complexity increases over time for both species, as expected, yet the fidelity in complexity remains constant. While there are some caveats that we have discussed (previously, and a bit more below), you presented an interesting and highly useful case-study for our field. As you rightly point out, little research has focused on selection of both models and their mimics. Thus, your paper will be valuable for broad audiences.

Overall, I think you did a nice job of addressing my concerns, while balancing the other concerns that were raised. My most major critique is that your fine-grained data does appear to reveal more intricate, cyclical, patterns. In your defense, you claim that these are driven by sample size, a claim I think would value from a bit more evidence (e.g., a test in a supplement). That said, Figure 1 does show the expected pattern and overall you have presented a complex topic neatly and succinctly.

I hope you haven't found my critiques overly critical. Likewise, I hope the comments were constructive and helpful. At this stage, the paper may require some minor tweaking but I do not feel I can provide any more feedback beyond what we discussed thus far. Good luck with your manuscript.

37Please find short responses and thoughts to this version (that I hope provide further assistance).

Best wishes,

Daniel

Dear Daniel,

Many thanks for your comments. We are glad you think the paper will be valuable for broad audiences and that you feel that your concerns were well-addressed. Below we respond to your remaining concerns.

Comments on the previous edits:

1. Two points

a. Differences in complexity do not relate to the actual patterns themselves. While I do understand that our field often uses two-tailed tests to analyze trait differences without giving too much thought to the traits themselves, in your case I do think it will pay to mention that prinia pay attention to the complexity (as a trait itself) rather than their paying attention to the complex trait. Perhaps some argument about neural processor, or simply a statement that we simply do not yet know everything they pay attention to. I do see one line early on, but the idea that hosts are responding to an abstract concept of 'complexity' deserves a bit more attention given its importance to your study.

Thank you for this suggestion. We have added information as you suggest; for instance the words “a synthetic measure of several pattern traits” in line 53, and “hosts perceive this measure of complexity” in line 54.

b. Direction of evolution between hosts and parasites: While I do understand that hosts were always more complex than parasites, they were not always evolving to be ever-more complex, while parasites followed in suite. Your new plot and data show apparent cyclical patterns, with a decrease in complexity in both host and parasite by the mid- to late-1980s, then a period of increase complexity in both by the 2010s, with a decline in parasite complexity by the end of that decade. If this was directional, my expectation would be that the prinia would be more complex, the parasite would get more complex,

38which would drive more complexity in the prinia, etc. I do see that overall, this is the case (see Figure 1), however your other plot from the response reveals a slightly more nuanced pattern.

Thank you very much for these thoughts and for the clear explanation. We agree that there are apparent patterns other than the overall one of increased complexity. You mention:

1. A decrease in complexity in the mid- to late-1980s.
2. An increase in complexity between the 1980s and early 2010s.
3. An apparent decrease in complexity in parasites in 2018–2020 (note that this isn't seen in hosts, which suggests that hosts and parasites are not changing symmetrically during these very short timescales).

To consider each of these in turn:

1. There is a lot of 'jumping around' in complexity in the mid-1980s – much more than in the 2010s. We note that each year has few eggs, usually <10 per species. Furthermore, the changes in complexity are evidently not statistically significant. Therefore, we would feel very unsure about making direct inferences about why in some years complexity in one or both species (apparently) increases or decreases.
2. Indeed, this seems very likely indicative of the overall pattern (as seen through various lines of evidence) that complexity increased over the full span of 50 years.
3. The apparent decrease in complexity in parasites in 2018–2020 does not appear to be statistically significant, and a visible decrease is only seen between 2019 and 2020.

Nevertheless, we acknowledge that there is not a monotonic increase in complexity over time, which in any case would not be expected given the variation in sample sizes (with <10 eggs in many years). The expected variation between years due to sample size (and/or other factors) in fact was one of the reasons to conduct further analyses (bootstrapping, dividing the dataset into the two main time periods etc.). Therefore, in the revised methods section, we point out that there were fluctuations in complexity, and that these may be attributed to several causes (Lines 208–218: "Median complexity appeared to fluctuate during certain periods (Extended Data Figure 1), such that the overall increase in complexity was not monotonic. These fluctuations are likely due to low sample sizes of eggs from specific years, combined with very high population-wide variation in complexity. Such sampling error is especially likely during periods such as the mid- to late-1980s, in which sample sizes were low for each year; correspondingly, fluctuations in complexity were apparent in these years. However, with these data we cannot rule out other selective pressures or environmental influences driving short-term increases or decreases in complexity in one or both species. Regardless of the cause of these apparent fluctuations,

they mean that we did not observe a monotonic increase in complexity in either species. Therefore, we conducted further analyses to test the robustness of the results of the linear model.”)

2. The evolution didn't seem to be very rapid: As pointed out above there seem to be consistent cyclical changes. Thanks for emphasizing the chase-away mechanism. I will not press you over how quickly evolution must proceed to be considered 'rapid'.

Many thanks for this comment. As discussed above, the fluctuations are on very short time-scales, so if they were products of selection then evolution would be exceedingly rapid! However, as also discussed above, the most likely interpretation is that these are simply artefacts of small sample sizes for some individual years. Rather, we considered the evolution to be 'rapid' since we observed changes over only 50 years despite hosts and parasites being vertebrates with correspondingly long generation times.

3. Two comments

a. Comments about imperfect mimicry: Great changes! Thank you.

b. Comments about how imperfect mimicry works on spots in this system: Also, great work.

Many thanks! We are glad you found the changes helpful.

4. Expectation of a linear relationship between year and complexity. Given that your data show no particular linear relationship with time (see the plot you inserted into the response letter, which was evident from your last submission), I still am quite perplexed why you expect a linear relationship between complexity and time in your dataset. It's clear that hosts are more complex than parasites, but complexity increases and decreases for both in more or less similar fashions. I cannot imagine how complexity would linearly relate to time (also since the dataset really has two time-frames, it may be questionable to do so). Your approach to handle divergent opinions on how to handle the linear analyses vs historical to modern comparisons, seems fair. Overall, your analysis seems OK. However, these data do suggest there's a few other interesting patterns.

Thanks for laying out these ideas. We used a linear model as the initial approach (constant parasitism pressure over time might be reasonably assumed to predict a linear change in complexity due to constant selection pressures) – and then, as you say, conducted additional tests of the robustness of the linear model which justified the linear approach. You point out that the dataset has two time-frames – this is an excellent point, and we dealt with this by testing for changes in complexity and mimetic fidelity

40over these two time-frames. You also point out the apparent finer-scale patterns in the data, which we have responded to under your comment 1b.

5. Alternatives: You have now provided some reasonable alternatives. Your point on red queen is fair enough, and well taken.

Again, many thanks for the suggestions!

Other minor comments:

A. Cyclical patterns in complexity among hosts and parasites rather than linear increases: I am not convinced (but easily could be) that the apparent cyclical patterns between hosts and parasites are a produce of sample size. Was sample size positively related to complexity? I do not disagree with this (true) statement “The reason that complexity should change over time is that hosts are more complex than prinias.” And while I do understand that chase away would suggest increasing complexity (linear increases) since the host was originally more complex.... Given the patterns that you have plotted, I do think it’s important to explain why they do not become more complex consistently over every sampled year. Instead, they change periodically and symmetrically in their degree of complexity. The reason I mentioned some global and environmental factors, which are known to impact some aspects of egg pigmentation, is that those factors could explain why both hosts and parasites would shift in tandem across these decades, but why hosts may retain their greater complexity. As both species live within the same environment and are subjected to the same pressures. One way or the other, it seems vital to explain why you expect the host and parasite to get more complex when it’s clear they simply change synchronously in complexity. If it is true that this is a statistical artifact you would need a test to demonstrate that sample size is related to complexity (so years with lower sample sizes have lower complexity because there are too few eggs). However, unfortunately, you would also need the accompanying test to show that the greater complexity in hosts is not only because you have greater sample size of host eggs (i.e., to show that those observations are not simply because of the sample size – I do not believe this is the case, see my next point).

Many thanks for the detailed comment here and for putting forward your argument that the patterns change periodically and symmetrically. Please see our response to your comment 1b, for why we do not feel that the patterns are cyclic or symmetrical. To add to our discussion there, we argued in the manuscript that we expect increases in complexity due prinias exhibiting higher complexity than cuckoo finches. There is no expectation for cuckoo finches to evolve lower complexity (away from prinias), nor

prinias to evolve lower complexity (making them more similar to cuckoo finches). We formed our hypotheses *a priori*, and therefore have retained these to ensure full transparency and to avoid post-hoc hypothesising – though as mentioned we have now added comment on the fluctuations between years.

Your suggestion about a possible cyclical pattern in fluctuations (though again, note that such fluctuations are only really apparent in years with low sample sizes, such as the mid- to late-1980s) is interesting, but without knowledge of what that pattern might be (i.e., how it might be parametrised), it is effectively impossible to model convincingly. Importantly, therefore, we tested whether the observed increase in complexity was *robust* to these fluctuations, and we found that our conclusions were indeed robust. Nevertheless, we agree that it is important to highlight that there was not a monotonic increase in complexity year after year (which we would not expect given that many years had <10 eggs), and so we have made the changes highlighted in our response to your comment 1b.

We see no reason to expect years with lower sample size to have lower complexity because complexity is measured per egg. The point is that the estimate of complexity is *uncertain* in years with low sample size (i.e., sampling error), not that complexity should be consistently lower (or higher) in these years. If we have a small sample size in one year and the eggs happen to have lower complexity, we would be uncertain whether that was due to the population having lower complexity that year, or because the few eggs sampled just happened to be those that had low complexity (i.e., that the sample median was lower than the population median). The same would be true for a year in which sample size was low and median complexity was high. And as Extended Data Figure 1 shows, the complexity in these years with low sample sizes varies a lot, such that differences between the years are clearly non-significant.

Overall, therefore, we respectfully disagree that we find ‘clear’ cyclic and synchronous patterns (though, as with any conceivable pattern, we cannot conclusively prove that such a pattern does not exist). Our analyses demonstrate the robustness of the conclusions that complexity increases over time regardless of years with low sample sizes or uncertain estimates of complexity.

B. I think the greater variance in the host would actually be a feature of this dynamic, but realize that this is a bit speculative. Great response though.

Many thanks. Indeed, previous work has discussed how variation in hosts is a feature of this dynamic (e.g. Spottiswoode and Stevens 2010 *Am Nat*, Lund et al., 2023 *Proc R Soc B*). We also now mention variation in the first section of the Methods when introducing the system (as well as the information in the Main Text that we included in the previous version at your suggestion).

C. Presentation on original line 75: Thanks for embracing this suggestion. I think the revised passage works nicely!

Many thanks for the initial suggestion! We are glad that it works well.

D. Logs and percentages: Thanks for clarifying. The presentation is still a little challenging, but you did a good job explaining why they are presented in this fashion. I apologize about my misleading comment about 0.04 log units not equaling 4%. Maybe I thought you used \log_{10} (I'm honestly not sure what I was thinking there, I'm sorry); however, while we do get 4%, if 0.04 is assigned to x in this equation, $(\exp(x)-1)*100$, this is increasingly untrue with greater values of x . Hence, my unease in seeing percentages to describe these values. Again, you did a fair job at explaining, but this will likely be a place that is hard to follow in your paper for most readers.

We appreciate this comment and your understanding of the clarification. As we discussed in our previous response, we understand the concern but feel that our presentation is the easiest to follow of the available alternatives, particularly as we explain in detail how the values were calculated in both the Main Text and Methods sections. Furthermore, percentage changes can be generated by calculating $\exp(x)$, where x is the estimate, regardless of the value of x . Thus, with greater values of x , $\exp(x)$ still gives us the correct percentage change.

E. Fading: I think your approach is fine. However, my concern is that the fading that occurs will impact how complexity gets quantified from your images as it impacts the relative contrast between spots and background coloration, among other features that relate to capturing and assessing complexity. So, while it may be true that for some morphs you would expect more contrast, this isn't going to be true across the board. There are strong nonlinear relationships with fading specific to each pigment, which occur through oxidation and photooxidation. That said, I think you've done all you can do and added reasonable citations and statements about the limitations.

Many thanks again for the advice and suggestions in the previous version, and your comment here. We are glad you find our explanations of the limitations clear, and the reasons why our results are conservative.

F. Estimation of the achromatic channel: Totally reasonable to use the green channel. I think the reasoning here is that it most closely matches the peak sensitivity of the double cone. My thinking was that for a standard image you may be better off with a normal grayscale image, which is calculated differently. Your rationale is reasonable and justified.

Thank you again for this comment and we are glad the clarification helps.

Reviewer #3 (Remarks to the Author):

Comments on 19066_2

Title: Chase-away evolution maintains imperfect mimicry despite rapid evolution of mimics

I found the authors addressed well with my comments, and thus I have no further comments.

In addition, by reading the authors' response to comments by other reviewers, I found the revision more focused.

Compared with molecular-based studies (e.g., genomes), time-consuming field studies would have few opportunities to get published in high-impact factor journals, and in turn, this would make more and more scientists lose their passion to work in the field.

For example, in recent years, there has not been a good trend that more and more papers were based on other papers' data (even such data was in problem), or just a meta-analysis using other studies' data. This would make research (in particular, ecological research) become a "bad" science.

In light of this, by using over 50 years' data, this study tracked the model and mimic phenotypes and showed that despite rapid evolution of parasites, there was no detectable increase in their mimetic fidelity to hosts, suggesting that the coevolutionary response in hosts was strong enough to prevent increases in mimetic fidelity, and the persistence of imperfect mimicry can be explained by chase-away evolution in models.

We are glad you found the manuscript interesting and improved! Thank you very much for your valuable input in the first round of review, and for your comments here about the field more broadly. We agree that field studies are important, and hope our study will contribute to encouraging researchers to conduct field-based studies of coevolution in natural systems.

Decision Letter, third revision:

22nd August 2023

Dear Dr. Dixit,

Thank you for submitting your revised manuscript "Chase-away evolution maintains imperfect mimicry despite rapid evolution of mimics." (NATECOLEVOL-221117968C). It has now been seen again by the original reviewers and their comments are below. The reviewers find that the paper has improved in revision, and therefore we'll be happy in principle to publish it in Nature Ecology & Evolution, pending minor revisions to satisfy the reviewers' final requests and to comply with our editorial and formatting guidelines.

[REDACTED]

Our ref: NATECOLEVOL-221117968C

4th September 2023

45Dear Dr. Dixit,

Thank you for your patience as we've prepared the guidelines for final submission of your Nature Ecology & Evolution manuscript, "Chase-away evolution maintains imperfect mimicry despite rapid evolution of mimics." (NATECOLEVOL-221117968C). Please carefully follow the step-by-step instructions provided in the attached file, and add a response in each row of the table to indicate the changes that you have made. Please also check and comment on any additional marked-up edits we have proposed within the text. Ensuring that each point is addressed will help to ensure that your revised manuscript can be swiftly handed over to our production team.

****We would like to start working on your revised paper, with all of the requested files and forms, as soon as possible (preferably within two weeks). Please get in contact with us immediately if you anticipate it taking more than two weeks to submit these revised files.****

In recognition of the time and expertise our reviewers provide to Nature Ecology & Evolution's editorial process, we would like to formally acknowledge their contribution to the external peer review of your manuscript entitled "Chase-away evolution maintains imperfect mimicry despite rapid evolution of mimics.". For those reviewers who give their assent, we will be publishing their names alongside the published article.

Nature Ecology & Evolution offers a Transparent Peer Review option for new original research manuscripts submitted after December 1st, 2019. As part of this initiative, we encourage our authors to support increased transparency into the peer review process by agreeing to have the reviewer comments, author rebuttal letters, and editorial decision letters published as a Supplementary item. When you submit your final files please clearly state in your cover letter whether or not you would like to participate in this initiative. Please note that failure to state your preference will result in delays in accepting your manuscript for publication.

Cover suggestions

We welcome submissions of artwork for consideration for our cover. For more information, please see our https://www.nature.com/documents/Nature_covers_author_guide.pdf target="new"> guide for cover artwork.

46Please submit your suggestions, clearly labeled, along with your final files. We'll be in touch if more information is needed.

Nature Ecology & Evolution has now transitioned to a unified Rights Collection system which will allow our Author Services team to quickly and easily collect the rights and permissions required to publish your work. Approximately 10 days after your paper is formally accepted, you will receive an email in providing you with a link to complete the grant of rights. If your paper is eligible for Open Access, our Author Services team will also be in touch regarding any additional information that may be required to arrange payment for your article.

Please note that *Nature Ecology & Evolution* is a Transformative Journal (TJ). Authors may publish their research with us through the traditional subscription access route or make their paper immediately open access through payment of an article-processing charge (APC). Authors will not be required to make a final decision about access to their article until it has been accepted. [Find out more about Transformative Journals](https://www.springernature.com/gp/open-research/transformative-journals)

Authors may need to take specific actions to achieve [compliance with funder and institutional open access mandates](https://www.springernature.com/gp/open-research/funding/policy-compliance-faqs). If your research is supported by a funder that requires immediate open access (e.g. according to [Plan S principles](https://www.springernature.com/gp/open-research/plan-s-compliance)) then you should select the gold OA route, and we will direct you to the compliant route where possible. For authors selecting the subscription publication route, the journal's standard licensing terms will need to be accepted, including [a self-archiving-and-license-to-publish](https://www.nature.com/nature-portfolio/editorial-policies/self-archiving-and-license-to-publish). Those licensing terms will supersede any other terms that the author or any third party may assert apply to any version of the manuscript.

[REDACTED]

[REDACTED]

Final Decision Letter:

19th September 2023

Dear Mr Dixit,

I am delighted to tell you that your manuscript (NATECOLEVOL-221117968D) has been accepted for publication in Nature Ecology & Evolution.

We have now transitioned to a unified Rights Collection system which will allow our Author Services team to quickly and easily collect the rights and permissions required to publish your work. Once your paper is typeset, you will receive an email with a link to choose the appropriate publishing options for your paper and our Author Services team will be in touch regarding any additional information that may be required.

Once this step is complete, your proof will be made available through our e.proofing system via a link in a forthcoming communication. The system will show you an HTML version of the article that you can correct online. We may make minor changes to enhance the lucidity of the text, and if necessary to make it conform to the journal's style. We therefore ask that you examine the proofs most carefully to ensure that we have not inadvertently altered the sense of your text in any way.

****Please note that you will not receive your proofs until the publishing agreement has been received.****

Due to the quick turn-around of this content, it is important that you let us know if you will be unreachable for the next 2-3 weeks after acceptance, in which case we ask that you kindly send us the contact information of someone who will be able to check the proofs and handle any last-minute problems. Please address any correspondence about your proofs to our Production team at rjsproduction@springernature.com.

Please note that we might decide to distribute a press release to news organizations worldwide. This is usually done about ten days before the paper is published and it could include details of your work. We are more than happy for your institution or funding agency to prepare its own press release, but it must mention the embargo date and Nature Ecology & Evolution. Please contact us for more details.

Acceptance is conditional on the manuscript's not being published elsewhere, and on there being no announcement of this work to the newspapers, magazines, radio or television before the publication date. Nature Ecology & Evolution, however, does allow the registered journalists who receive our press release to have copies of papers a week before publication under strict embargo conditions, solely for the purpose of publicizing the work in the media. We permit these journalists to show papers to independent specialists a few days in advance of publication, again under embargo conditions, solely for the purpose of commenting on the work described. These restrictions are not intended to deter you from presenting your data at academic meetings and conferences, but any inquiries from the media

48about the papers not yet scheduled for publication should be referred to us.

[REDACTED]